# Grid-Stretching Capability for the GEOS-Chem 13.0.0 Atmospheric Chemistry Model

Liam Bindle[1,2], Randall V. Martin[1,2,3], Matthew J. Cooper[2,1], Elizabeth W. Lundgren[4],
Sebastian D. Eastham[5], Benjamin M. Auer[6], Thomas L. Clune[6], Hongjian Weng[7], Jintai Lin[7],
Lee T. Murray[8], Jun Meng[2,1,9], Christoph A. Keller[6,10], William M. Putman[6], Steven Pawson[6], and
Daniel J. Jacob[4]

[1]Department of Energy, Environmental and Chemical Engineering, Washington University, St. Louis, Missouri, USA
[2]Department of Physics and Atmospheric Science, Dalhousie University, Halifax, NS, Canada
[3]Harvard-Smithsonian Centre for Astrophysics, Cambridge, MA, USA
[4]John A. Paulson School of Engineering and Applied Sciences, Harvard University, Cambridge, MA, USA
[5]Laboratory for Aviation and the Environment, Massachusetts Institute of Technology, Cambridge, MA, USA
[6]Global Modeling and Assimilation Office, NASA Goddard Space Flight Center, Greenbelt, MD, USA
[7]Laboratory for Climate and Ocean–Atmosphere Studies, Department of Atmospheric and Oceanic Sciences, School of
Physics, Peking University, Beijing, China
[8]Department of Earth and Environmental Sciences, University of Rochester, Rochester, NY, USA
[9]Atmospheric and Oceanic Sciences, University of California, Los Angeles, Los Angeles, California, USA
[10]Universities Space Research Association, Columbia, MD, USA

**Correspondence:** Liam Bindle (liam.bindle@wustl.edu)

**Abstract.** Modeling atmospheric chemistry at fine resolution globally is computationally expensive; the capability to focus on specific geographic regions using a multiscale grid is desirable. Here, we develop, validate, and demonstrate stretched grids in the GEOS-Chem atmospheric chemistry model in its high-performance implementation (GCHP). These multiscale grids are specified at runtime by four parameters that offer users nimble control of the region that is refined and the resolution of the refinement. We validate the stretched-grid simulation versus global cubed-sphere simulations. We demonstrate the operation and flexibility of stretched-grid simulations with two case studies that compare simulated tropospheric $NO_2$ column densities from stretched-grid and cubed-sphere simulations to retrieved column densities from the TROPOspheric Monitoring Instrument (TROPOMI). The first case study uses a stretched grid with a broad refinement covering the contiguous US to produce simulated columns that perform similarly to a C180 ($\sim$50 km) cubed-sphere simulation at less than one-ninth the computational expense. The second case study experiments with a large stretch-factor for a global stretched-grid simulation with a highly localized refinement with $\sim$10 km resolution for California. We find that the refinement improves spatial agreement with TROPOMI columns compared to a C90 cubed-sphere simulation of comparable computational demands. Overall we find that stretched grids in GEOS-Chem are a practical tool for fine resolution regional- or continental-scale simulations of atmospheric chemistry. Stretched grids are available in GEOS-Chem version 13.0.0.

## 1 Introduction

Global simulations of atmospheric chemistry are computationally demanding. Chemical mechanisms in the troposphere typically involve more than 100 chemical species, emitted by anthropogenic and natural sources, with production and loss by chemical reactions, and mixing through 3-D transport on all scales. Typical global model resolutions are on the order of hundreds of kilometers and generally limited by the degree of model parallelism. Massively parallel models such as GEOS-Chem in its high-performance implementation (GCHP; Eastham et al. (2018)) can run on more than 1000 cores (Zhuang et al., 2020) with demonstrated capability of 50 km resolution. The coarse resolution of global models can lead to systematic errors in applications when scales of variability finer than the model resolution are relevant, such as vertical transport and scavenging by convective updrafts (Mari et al., 2000; Li et al., 2018, 2019), nonlinear chemistry such as $NO_x$ titration (Valin et al., 2011), localized emission sources (Davis et al., 2001; Freitas et al., 2007), a priori profiles for satellite retrievals (Heckel et al., 2011; Goldberg et al., 2017; Kim et al., 2018), and simulated concentrations for population exposure estimates (Punger and West, 2013; Li et al., 2016). Grid refinement is commonly used for simulations that need to capture fine-scale modes of variability. Here we implement grid refinement in GCHP using a technique that stretches the model grid to enhance its resolution in a user-defined region, enabling massively parallel global multiscale simulations of atmospheric chemistry. We validate the implementation, discuss key considerations for stretched-grid simulations, and demonstrate the capability.

The general approaches to grid refinement are nesting, adaptive grids, and grid-stretching. Nesting broadly describes the use of a coarse global grid in combination with finer regional grids. In one-way nesting, a coarse global simulation generates boundary conditions for one or more regional simulations (Miyakoda and Rosati, 1977; Wang et al., 2004; Lin et al., 2020). One-way nesting is simple yet effective when regional feedbacks on the global simulation are not required. In two-way nesting, the coarse global simulation is dynamically coupled to the regional simulations to allow for feedbacks (Zhang et al., 1986; Krol et al., 2005; Yan et al., 2014; Zängl et al., 2015; Feng et al., 2020). Two-way nesting is more complex, technically, than one-way nesting, but the regional feedbacks from finely resolved chemical and physical processes can improve global simulations of trace gases such as CO and $O_3$ (Yan et al., 2014, 2016). One-way and two-way nesting are supported by the single-node version of GEOS-Chem "Classic" (Wang et al., 2004; Yan et al., 2014), and by GEOS-Chem using Weather Research and Forecasting (WRF) meteorology (WRF-GC; Lin et al. (2020); Feng et al. (2020)). An adaptive grid is a grid that supports dynamic refinement. Adaptive grids allow the refinement to continuously target regions where simulation accuracy is the most sensitive to model resolution (Tomlin et al., 1997; Slingo et al., 2009; Garcia-Menendez and Odman, 2011). Compared to static refinements, adaptive grids can better capture strong chemical gradients, which can improve the representation of nonlinear chemistry and reduce numerical diffusion (Srivastava et al., 2000; Garcia-Menendez et al., 2010). Adaptive grids are promising, but they are inherently complex and have not yet been used for global atmospheric chemistry simulations. Stretched grids are global grids that are "stretched" by an analytic transform to enhance the grid resolution in a region (Courtier and Geleyn, 1988; Krinner et al., 1997; Fox-Rabinovitz et al., 2006, 2008; McGregor and Dix, 2008; Tomita, 2008; Harris et al., 2016; Uchida et al., 2016). This stretching creates a global grid with a single refinement and smooth gradual changes in resolution. A key advantage of grid-stretching is simplicity. Stretching does not change the logical structure (topology) of the grid, so

fundamental changes to the model structure are not required and two-way coupling is inherent. An additional benefit is that
lateral boundary conditions are not required. A drawback of grid-stretching, however, is that a stretched-grid simulation has a
single refinement. The first use of stretched grids for atmospheric chemistry simulations was in Allen et al. (2000) and Park
et al. (2004); more recently, Goto et al. (2015) used a stretched grid for a fine-resolution simulation of aerosols in Japan, and
Trieu et al. (2017) used a stretched grid for a fine-resolution simulation of surface-level $O_3$ in Japan.

Several recent works set the stage for the development of grid-stretching in GEOS-Chem. Long et al. (2015) developed the
grid-independent capability of GEOS-Chem. Harris et al. (2016) developed the stretched grid capability for the GFDL Finite-
Volume Cubed-Sphere Dynamical Core (FV3), which is used to calculate advection in GCHP. Eastham et al. (2018) developed
the capability for GEOS-Chem to operate on cubed-sphere grids in a distributed memory framework for massive paralleliza-
tion, and to use the Model Analysis and Prediction Layer (MAPL; Suarez et al. (2007)) of the NASA Global Modeling and
Assimilation Office (GMAO) together with the Earth System Modeling Framework (ESMF; Hill et al. (2004)) to couple model
components. GEOS-Chem version 12.5.0 added grid-independent emissions that produce consistent emissions regardless of
the model grid (Weng et al., 2020; The International GEOS-Chem User Community, 2019). Most recently, MAPL version 2
(Thompson et al., 2020) of the NASA GMAO added stretched grid support.

The remainder of the manuscript is organized as follows. Sect. 2.1 describes GCHP and the gnomonic cubed-sphere model
grid. Sect. 2.2 describes the stretching transform, which is based on the Schmidt (1977) transform and follows the methodology
of Harris et al. (2016). Sect. 2.3 discusses considerations for stretched-grid simulations and a simple procedure for choosing
an appropriate stretch-factor. Sect. 2.4 describes the shared model configuration that is used for all of the simulations in this
manuscript. Sect. 2.5 validates the stretched-grid implementation by a comparison of simulated oxidants and $PM_{2.5}$ concentra-
tions from stretched-grid and cubed-sphere simulations. Finally, case studies in Sect. 3.1 and Sect. 3.2 demonstrate and explore
grid-stretching in GCHP by considering the application of comparing simulated $NO_2$ columns with regional observations from
TROPOspheric Monitoring Instrument (TROPOMI).

## 2 Development of Stretched Grids in GEOS-Chem

### 2.1 GEOS-Chem in its High-Performance Implementation (GCHP)

We use GEOS-Chem version 13.0.0 in its high-performance implementation (GCHP; Eastham et al. (2018)). GEOS-Chem,
originally described in Bey et al. (2001), simulates tropospheric-stratospheric chemistry by solving 3-D chemical continuity
equations. GCHP uses MAPL (Suarez et al., 2007) and ESMF (Hill et al., 2004) to facilitate the coupling of model components
and use of the High-Performance Computing (HPC) infrastructure. The 3-D advection component is the GFDL Finite-Volume
Cubed-Sphere Dynamical Core (Putman and Lin, 2007). Columnar operators (Long et al., 2015) are used for columnar or
local calculations such as convection and chemical kinetics. Emissions are aggregated, parameterized, and computed with the
Harmonized Emissions Component (HEMCO) described in Keller et al. (2014). Stratospheric chemistry is simulated using the
Unified Chemistry Extension (UCX) described in Eastham et al. (2014). Offline meteorological data are from the Goddard
Earth Observing System (GEOS) data assimilation system (Rienecker et al., 2008; Todling and El Akkraoui, 2018). Emissions

**Table 1.** Characteristics of various cubed-sphere grids and global latitude-longitude grids.

| | # of grid-boxes per level[1] | Resolution[2] | |
| --- | --- | --- | --- |
| | | Average | Range |
| Cubed-sphere grids | | | |
| C24 | 3,456 | 384 km | 310–461 km |
| C48 | 13,824 | 192 km | 153–231 km |
| C60 | 21,600 | 154 km | 123–185 km |
| C90 | 48,600 | 102 km | 82–123 km |
| C180 | 194,400 | 51 km | 41–62 km |
| C360 | 777,600 | 26 km | 20–31 km |
| Regular lat-lon grids | | | |
| 4°×5° | 3,600 | 376 km | 88–472 km |
| 2°×2.5° | 12,960 | 198 km | 33–249 km |
| 1°×1° | 64,800 | 89 km | 10–111 km |
| 0.5°×0.625° | 207,360 | 50 km | 4–62 km |
| 0.25°×0.3125° | 829,440 | 25 km | 2–31 km |

1. Number (#) of grid-boxes is listed for one vertical level. The GEOS-FP and MERRA-2 meteorological fields currently have 72 vertical levels.

2. Here we define the resolution of a grid-box as the square root of its area.

and meteorological input data are regridded online by ESMF using the first-order conservative scheme originally described in Ramshaw (1985). GCHP discretizes the atmosphere with a gnomonic cubed-sphere grid with levels extending from the surface to 1 Pa. The cubed-sphere grid has several advantages over a regular latitude-longitude grid, stemming from its more uniform grid-boxes that benefit the parallelization and numerical stability of transport (Eastham et al., 2018). The horizontal resolution of a GCHP simulation is a key determinant of its computational demands.

An example of the horizontal grid is illustrated in Figure 1. A gnomonic cubed-sphere grid is a mosaic of six grids hereafter referred to as *faces*. Each face is a logically square grid that is regularly spaced in a gnomonic projection centered on the face. One of the six faces is highlighted in Figure 1. The position of the faces are fixed, and the center of the first face is 0 °N, 10 °W. Hereafter, we refer to a gnomonic cubed-sphere grid as simply a *cubed-sphere* or a *cubed-sphere grid*. The horizontal resolution of a cubed-sphere is dictated by its size, which is an even integer denoted with the notation C$N$ (e.g., C180); each face in the 6-face mosaic is an $N \times N$ grid. The computational demand of a GCHP simulation is proportional to the total number of grid-boxes. Table 1 provides a comparison of cubed-sphere grids and conventional latitude-longitude grids. We note for context that GEOS-Chem Classic can use a 2°×2.5° or 4°×5° global grid.

Offline meteorological data for GCHP are provided by the GMAO. GCHP uses a local archive of the GEOS-FP or MERRA-2 data product. GEOS-FP is a near real-time analysis product with a 0.25°×0.3125° grid. MERRA-2 is a reanalysis product with a 0.5°×0.625° grid. The effect of topography and surface type on transport is implicit in the meteorological data that drives transport. Both data products have a 72-level terrain-following hybrid-sigma pressure grid that extends from the surface to 1 Pa. GCHP uses a vertical grid that is identical to the meteorological data, so vertical regridding is not required.

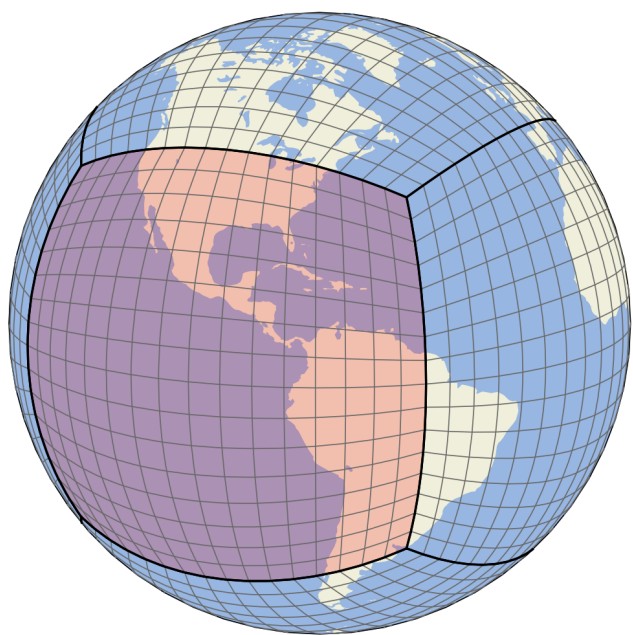

**Figure 1.** Illustration of a C16 grid. A cubed-sphere grid always has six faces. One of the six faces is highlighted for illustrative purposes. The highlighted face is a 16×16 grid that is regularly spaced in a gnomonic projection centered on the face.

## 2.2 Grid-stretching procedure

The grid-stretching procedure in GCHP uses a simplified form of the Schmidt (1977) transform for gnomonic cubed-sphere grids, following the methodology of Harris et al. (2016). The Schmidt transform can be applied to any grid, and effectively stretches the grid to increase its density in a region. The procedure has two steps, starting with a standard cubed-sphere grid. First, the grid is refined at the South Pole by remapping the grid coordinate latitudes with a modified Schmidt transform

$$\phi'(\phi) = \arcsin \frac{D + \sin\phi}{1 + D\sin\phi} \quad \text{where} \quad D = \frac{1 - S^2}{1 + S^2} \tag{1}$$

where $\phi$ is an input latitude, $\phi'$ is the output latitude, and $S$ is a parameter called the *stretch-factor*. The stretch-factor controls the strength of the operation, which effectively attracts the grid coordinates towards the South Pole along meridians. The second step is rotating the entire grid so that the refinement is repositioned to the desired region. The user specifies a *target latitude*, $T_\phi$, and *target longitude*, $T_\theta$. The refinement is re-centered to these coordinates by rotating the grid.

Figure 2 illustrates the effect of $S$ on stretching a cubed-sphere grid. A stretch-factor greater than one causes stretching. Larger stretch-factors cause more stretching, and result in a finer and more localized refinement. The resolution at the center of the refinement is approximately $S$ times finer than it was before stretching, and similarly, the antipode resolution is approximately $S$ times coarser. These relative changes are approximate since the Schmidt transform is continuous and the grid-boxes have nonzero length edges. The grids in Figure 2 illustrate three noteworthy features of stretched-grids: (1) the changes in

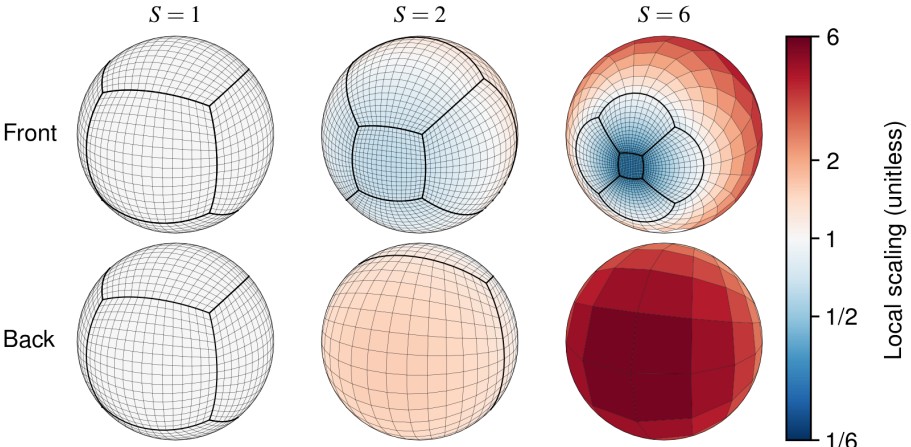

**Figure 2.** Three stretched-grids that illustrate the effect of the stretch-factor ($S$) on stretching a C16 cubed-sphere. Local scaling is the relative change to a grid-box's edge length induced by stretching.

resolution are smooth, (2) the refined domain diminishes as $S$ increases, and (3) grid-boxes outside the refined domain expand. The cubed-sphere face at the center of the refined domain is called the *target-face*.

The relative change to a grid-box size from stretching can be quantified by *local scaling*. This quantity represents the effect of grid-stretching at a given point. For a stretch-factor of $S$, the local scaling at a given point depends exclusively on how far that point is from the target coordinate. Local scaling, $L$, can be derived from Eqn. 1, and expressed as

$$L(\Theta; S) = \frac{1 + \cos\Theta + S^2(1 - \cos\Theta)}{2S} \tag{2}$$

where $\Theta$ is the angular distance to the target point. Appendix B contains the derivation of Eqn. 2. Figure 3 shows local scaling as a function of distance, for stretch-factors between 1 and 10. Overlaid are dashed lines that show the lateral position of the face edges after stretching. In the target-face, local scaling is approximately $1/S$ and nearly constant. The grid is refined up to the distance where $L = 1$ and coarsened at farther distances.

Four scalar parameters fully describe a stretched-grid: the size of the cubed-sphere, the stretch-factor ($S$), the target latitude ($T_\phi$), and the target longitude ($T_\theta$). These concise parameters are conceptually simple, precise, and give the user nimble control of the grid. The combination of cubed-sphere size and stretch-factor controls the grid resolution. The target latitude and longitude specify the center of the refined domain. Moderate stretch-factors (e.g., 1.4–3.0) are suitable for broad refinements for continental-scale studies. Large stretch-factors (e.g., >5.0) are suitable for localized refinements for regional-scale studies.

## 2.3 Choosing an appropriate stretch-factor

Although the stretch-factor is a well-defined parameter, appropriate values for atmospheric chemistry simulations will be application-specific and moderately variable. For computational efficiency, it is desirable to use the largest viable stretch-factor, to achieve the finest refinement for a given cubed-sphere size. However, larger stretch-factors also result in a smaller

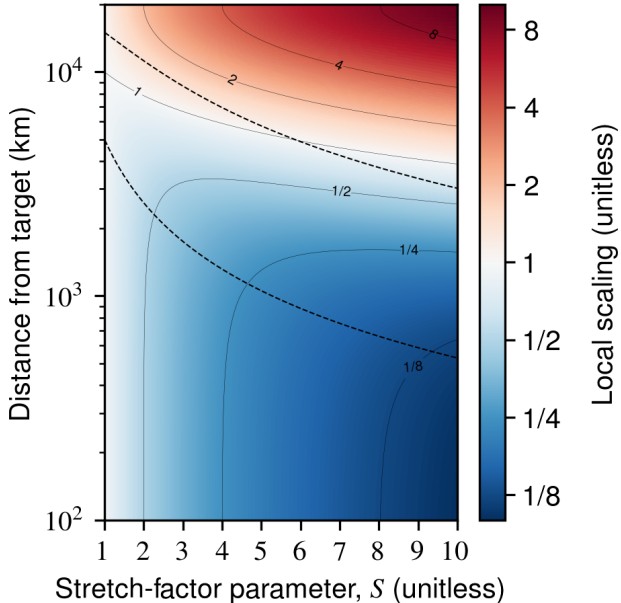

**Figure 3.** Local scaling as a function of distance from the target point, for stretch-factors in the range 1–10. The dashed lines show the distance from the target point to the cubed-sphere face edges after stretching. The lower dashed line is the distance to the center of the target-face's edges, and the upper dashed line is the distance to the center of the opposite face edges.

refinement and in coarser resolutions outside the refined domain. To determine the maximum suitable stretch-factor for a given
application, one should consider the size of the domain that should be refined, and the sensitivity of the study species to coarse resolutions outside the refinement. A simple procedure for choosing a stretch-factor, $S$, is choosing the maximum stretch-factor subject to two constraints:

1. **Constraining $S$ by the size of the refined domain.** Local scaling is approximately constant and equal to $1/S$ throughout the target-face. Therefore, defining the refined domain as the region where resolution is enhanced by a factor of $\sim S$, the
target-face is a reasonable approximation of the refined domain. For a target-face with width greater than $\mathrm{w_{tf}}$, $S$ must satisfy

$$S \leq 0.414 \, \cot\left(\mathrm{w_{tf}}/4 \, \mathrm{r_E}\right) \tag{3}$$

where $\mathrm{r_E}$ is Earth's radius. Here, we define target-face width, $\mathrm{w_{tf}}$, as the edge-to-edge distance across the center of the face. For example, for a refined domain with a diameter of at least 3000 km, the target-face should be at least 3000 km
wide, so the stretch-factor should be less than or equal to 3.5. Alternatively, one could inspect Figure 3 to find the distance from the target point to the edge of the target-face for a given value of $S$.

2. **Constraining $S$ by a maximum and minimum resolution.** The target resolution is $S^2$ times finer than its antipode's resolution; therefore, a constraint for $S$ based on the desired maximum resolution, $\mathrm{R_{max}}$, and minimum resolution, $\mathrm{R_{min}}$,

is

$$S \leq \sqrt{\mathrm{R_{max}/R_{min}}} \tag{4}$$

The minimum resolution, $\mathrm{R_{min}}$, is an important consideration. $\mathrm{R_{min}}$ imposes a limit on the coarsest resolution outside of the refined domain. Therefore, the choice of an appropriately fine $\mathrm{R_{min}}$ can reduce potential bias in species in the coarse grid outside of the refined domain from affecting the study species in the refined domain. For example, for a maximum and minimum resolution comparable to C360 and C24 cubed-spheres, $S \leq 3.9$, or for a maximum and minimum resolution comparable to C360 and C48 cubed-spheres, $S \leq 2.7$

Once the constraints for $S$ are determined, one can choose $S$ and the grid size for their simulation. For example, for a stretched-grid with a refined domain with a diameter greater than 3000 km, a maximum resolution of C360, and a minimum resolution of C24, the constraints would be $S \leq 3.5$ and $S \leq 3.9$. To determine the grid size and stretch-factor for a simulation from these constraints, first, assume $S = 3.5$ as an initial value. For a maximum resolution comparable to a C360 grid with $S = 3.5$, the grid size would be C102.9 ($360/3.5 = 102.9$). Since the grid size must be an even integer, one should round up to C104 and choose $S = 3.46$ ($360/104 = 3.46$).

The single refinement and the expansion of grid-boxes outside the refined domain are important limitations of grid-stretching. The coarse grid outside the refined domain is susceptible to resolution-dependent biases in $O_3$ and CO (Wild and Prather, 2006; Yan et al., 2014, 2016), which could influence the representation of chemical processes in the refined domain. Therefore, the choice of an appropriately fine minimum resolution is important ($\mathrm{R_{min}}$ in constraint 2). Generally, stretched-grid simulations are well suited for applications that are principally sensitive to emissions and physical processes in the refined domain. For example, stretched-grid simulations are well suited for regional studies of boundary layer concentrations of short-lived species. Applications such as evaluations of the global tropospheric $O_3$ budget are better suited for standard cubed-sphere simulations.

## 2.4 Model configuration

The simulations in this manuscript use a shared model configuration of GCHP version 13.0.0-alpha.3. We use default emissions for all species. Table 2 summarizes the emission inventories that represent sources of $NO_x$ in the simulations. Briefly, we use monthly anthropogenic emissions based on the National Emission Inventory (NEI) for 2011 with updated annual scaling factors to account for changes in annual totals since 2011. In 2019, the scaling factor for NO was 0.61. Biomass burning emissions are from the Global Fire Emissions Database version 4 (GFED4; van der Werf et al. (2017)). Aircraft emissions are from the Aviation Emissions Inventory Code (AEIC; Stettler et al. (2011)). We use inventories for $NO_x$ emissions from soil microbial activity and lightning calculated offline at the native resolution of the meteorology (Weng et al., 2020; The International GEOS-Chem User Community, 2019). Offline soil $NO_x$ emissions are based on the scheme described in Hudman et al. (2012), and offline lightning $NO_x$ emissions are based on the scheme described in Murray et al. (2012). For meteorological data, we use GEOS-FP data because it offers superior horizontal resolution data for advection input variables ($0.25° \times 0.3125°$). All simulations use a 10-minute timestep for chemistry and a 5-minute timestep for transport.

**Table 2.** Emissions inventories in GEOS-Chem 13.0.0 that represent $NO_x$ sources in the simulations conducted in this manuscript.

| Source type | Inventory | Resolution | Reference/Notes |
|---|---|---|---|
| Anthropogenic | | | |
|   US | NEI-2011 | $0.1°\times0.1°$ | Annual totals are updated for 2018/19[1] |
|   Asia | MIX | $0.25°\times0.25°$ | Li et al. (2017) |
|   Global | CEDS[2] | $0.5°\times0.5°$ | McDuffie et al. (2020) |
| Lightning | — | $0.25°\times0.3125°$ | Murray et al. (2012) |
| Soil | — | $0.25°\times0.3125°$ | Weng et al. (2020) |
| Biomass burning | GFED4 | $0.25°\times0.25°$ | van der Werf et al. (2017) |
| Aircraft | AEIC | $1.0°\times1.0°$ | Stettler et al. (2011) |
| Shipping | CEDS | $0.5°\times0.5°$ | McDuffie et al. (2020) |

1. Monthly fluxes, including diurnal and weekday–weekend variations and vertical allocations, based on criteria pollutants National Tier 1 from https://www.epa.gov/air-emissions-inventories/air-pollutant-emissions-trends-data (Accessed May 8, 2020).

2. Community Emissions Data System (CEDS)

Simulations are named according to their grid. Standard cubed-sphere simulations are named by their resolution with the suffix "-global", referring to their resolution being quasi-uniform globally. For example, a standard C180 cubed-sphere simulation is named "C180-global". Stretched-grid simulations are named according to their refinement effective resolution, with a suffix denoting the region that is refined. For example, a stretched-grid simulation with an effective resolution of C180 in the contiguous US is named "C180e-US". "C180e" refers to the stretched-grid refinement being comparable to the resolution of a C180 cubed-sphere grid.

## 2.5    Validating the stretched-grid capability

Next, we test the implementation of stretched-grid by comparing the concentrations of oxidants and $PM_{2.5}$ from stretched-grid and cubed-sphere simulations. Prior tests were also conducted, to identify and fix several technical errors in some component capabilities. We choose the contiguous US as the domain for this comparison and a standard C96 cubed-sphere simulation to serve as the control simulation (C96-global). The stretched-grid simulation, C96e-NA, has a grid size of C48 and stretching parameters $S = 2.4$, $T_\phi = 35°$ N, and $T_\theta = 96°$ W. The target point was chosen so the target-face approximately encompassed the populous regions of North America. The stretch-factor was chosen so that the average resolution of C96e-NA was equal to the average resolution of C96-global in the contiguous US; we note that the stretch-factor is 2.4, rather than 2.0, because the US is a region where the standard cubed-sphere grid has a finer resolution than its global average as shown in Appendix A. Figure 4 compares the resolution of C96e-NA and C96-global grids.

A consequence of the similar resolution of the C96-global and C94-globals grids is that their grid-boxes have little overlap; this makes their comparison sensitive to the precision of upscaling emissions (aliasing effects, i.e., differences in upscaled emissions like NO point sources from differences in how the grids cover a region). To calibrate an expectation for these differences, we compare C96-global to a second standard cubed-sphere simulation, C94-global. We choose C94-global because the similar resolution minimizes the frequency that its grid-boxes overlap with the C96-global grid-boxes. Therefore, the

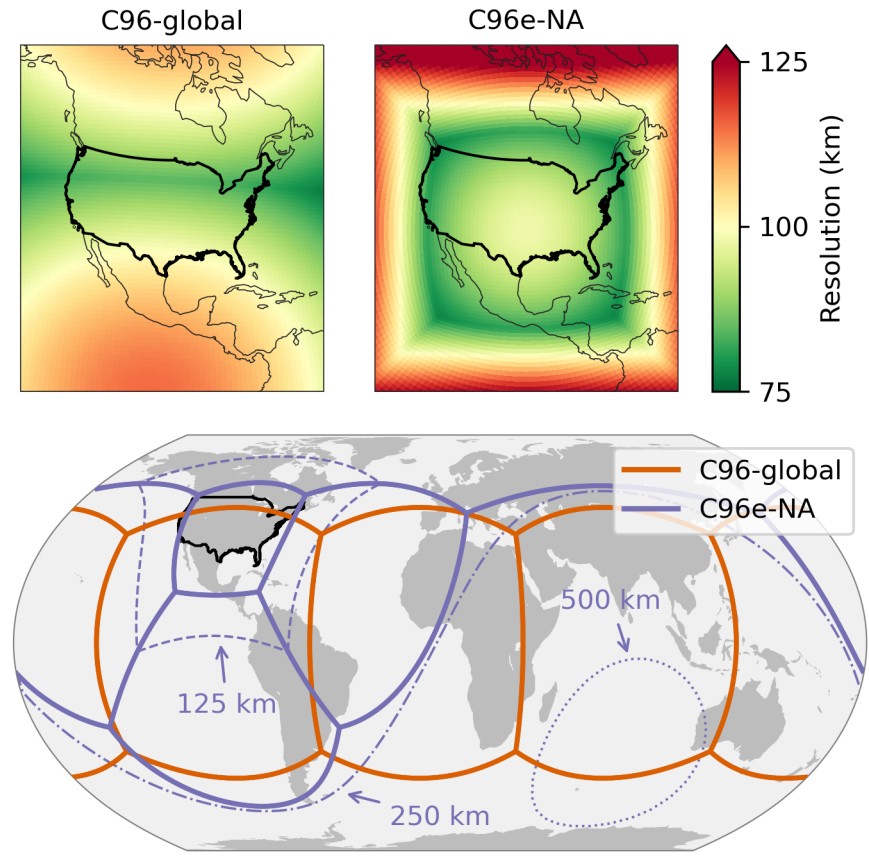

**Figure 4.** Comparison of C96-global and C96e-NA grids. The top panels show the variability of resolution for both grids. The bottom panel shows the grid face edges for C96-global and C96e-NA, with the 125 km, 250 km, and 500 km resolution contours for C96e-NA.

comparison of C94-global and C96-global isolates differences from aliasing effects. The frequency of grid-box overlap between C96-global and C94-global is analogous to a beat frequency of 2; the regions where overlap occurs is near the edges and across the center lines of the faces.

Figure 5 compares C96e-NA with C96-global, and C94-global with C96-global. These comparisons are for the fourth simulation month, to accommodate relaxation time for CO and $O_3$ (the simulations started June 1, 2018 and ran through October 1, 2018). All three simulations used an identical model configuration, apart from the grid, which is described in Sect. 2.4. The scatter near the surface is caused by differences in the spatial allocation of emissions on different grids (the precision of upscaling). Figure 5 shows that simulated averages from C96e-NA are consistent with C96-global and that their differences are

comparable to the differences between two similar cubed-sphere simulations. We note a small low bias in ozone above 400 hPa associated with nonlinear chemical sensitivity to coarse grid-boxes outside of the refined region. The C96e-NA grid resolution is coarser than 250 km in most of Asia, while the C96-global grid is finer than 125 km (Figure 4). A finer grid size or a smaller

stretch-factor could be used to increase resolution outside the refinement if this type of bias is of concern for a stretched-grid application.

## 3   Stretched-Grid Case-Studies

Next, we demonstrate stretched-grid simulations with GCHP by conducting case studies in Sect. 3.1 and 3.2. The applications we consider are regional comparisons of simulated tropospheric $NO_2$ columns with observations. $NO_2$ is chosen because it is a well-measured species, and the sensitivity of its simulated concentrations to model resolution and local chemical and physical processes make it a prime example of an application that is well-suited for a stretched-grid simulation. Here we explore two of the primary reasons one might use the stretched-grid capability: (1) for a computationally efficient regional simulation, and (2) to realize a finer resolution than otherwise possible. Sect. 3.1 considers a comparison of columns in the contiguous US; columns are simulated at ∼50 km resolution, with stretched-grid and standard cubed-sphere simulations, to examine the ability of a stretched-grid simulation to produce similar results to a cubed-sphere simulation at a lesser computational expense. Sect. 3.2 experiments with the use of a very large stretch-factor, for a simulation targeting California with ∼10 km resolution and modest computational requirements.

We compare simulated tropospheric $NO_2$ column densities to retrieved column densities from TROPOMI. TROPOMI is an instrument onboard the Sentinel-5P satellite launched in 2018 into a sun-synchronous orbit with a local overpass time of 13:30. TROPOMI includes ultraviolet and visible band spectrometers. The retrieved $NO_2$ column densities have $3.5 \times 5.5$ km$^2$ resolution and are calculated using a modified version of the Dutch OMI $NO_2$ (DOMINO) retrieval algorithm (Boersma et al., 2011, 2018). We include observations with retrieved cloud fractions less than 10 %. Retrieved $NO_2$ column densities are sensitive to the a priori profiles used to calculate the air mass factors (Boersma et al., 2018; Lorente et al., 2017). To avoid spurious differences from the retrieval a priori profiles when comparing simulated and retrieved $NO_2$ column densities, we recalculate the air mass factors with the mean simulated relative vertical profiles (shape factors) from the stretched-grid simulations following the approach described in Cooper et al. (2020) and Palmer et al. (2001). Evaluations of TROPOMI $NO_2$ columns show good correlation with ground-based measurements with a small low bias (Griffin et al., 2019; Ialongo et al., 2020; Zhao et al., 2020; Tack et al., 2020).

### 3.1   A stretched-grid simulation with a moderate stretch-factor

Here we consider a comparison of simulated and observed $NO_2$ columns in the contiguous US. We compare simulated columns with ∼50 km resolution from stretched-grid and standard cubed-sphere simulations with observations from TROPOMI. Our focus is on the technical ability to use a stretched-grid for an efficient fine-resolution simulation of the columns, compared to using a standard cubed-sphere for this purpose. The contiguous US is a large refinement domain, so a moderate stretch-factor is appropriate. This case is intended to be representative of a typical case where grid refinement is used in global models.

For the cubed-sphere simulation, we use a C180 grid and refer to it as C180-global. For the stretched-grid simulation, we use a stretch-factor of $S = 3.0$, a grid size of C60, and a target point 36° N, 99° W. We refer to the stretched-grid simulation as

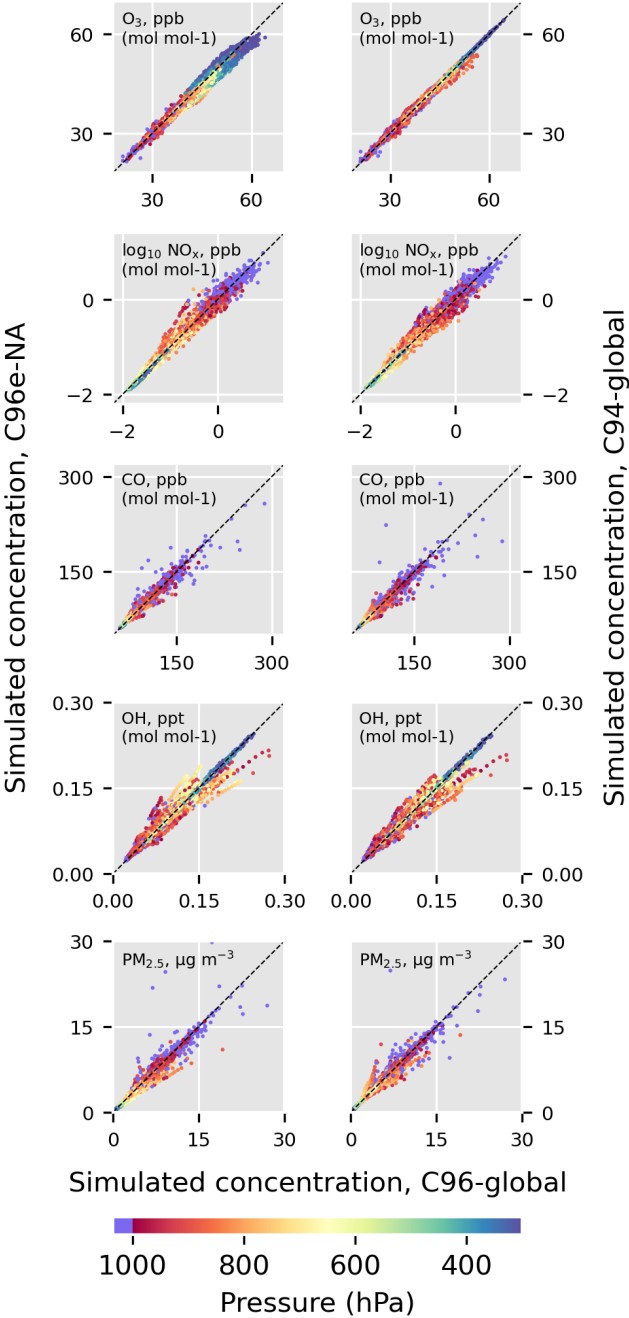

**Figure 5.** Comparison of simulated concentrations from C96e-NA with C96-global, and C94-global with C96-global. Points are 1-month time averages for each grid-box in the contiguous US for the fourth simulation month (September 2018). The right column gauges expected differences due to the precision of upscaling emissions to different grids. Concentrations in the lowermost model level are shown in purple.

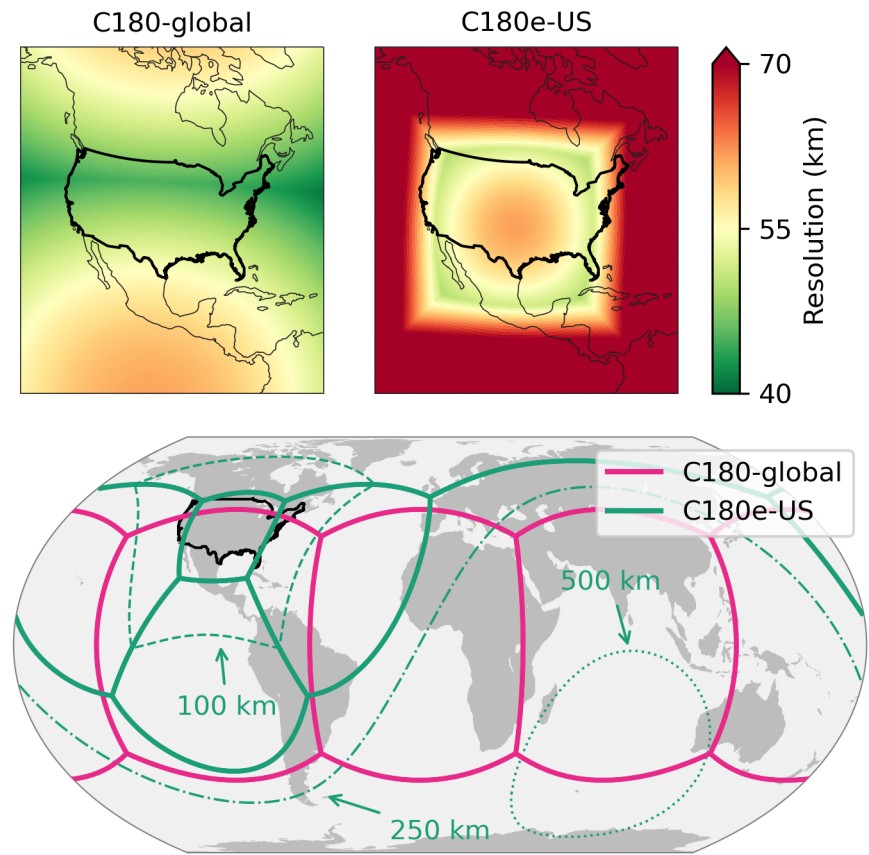

**Figure 6.** Comparison of C180-global and C180e-US grids. The top panels show the variability of resolution for both grids. The bottom panel shows the grid face edges for C180-global and C180e-US, with the 100 km, 250 km, and 500 km resolution contours for C180e-US.

C180e-US. The stretching parameters are chosen so that the target face approximately encompasses the contiguous US. Figure 6 shows the resolution of the two simulation grids. In the contiguous US, the average resolution of C180-global is 47.9 km, and the average resolution of C180e-US is 56.6 km. The total number of grid-boxes in C180e-US is 9 times fewer than the number of grid-boxes in C180-global (c.f. C60 and C180 grids in Table 1). Both simulations use an identical model configuration, apart from the grid. The model configuration is described in Sect. 2.4. To examine C180e-US without grid-stretching, we conduct a
third simulation, C60-global.

  Figure 7 shows tropospheric $NO_2$ column densities from TROPOMI, C180-global, C180e-US, and C60-global for the US in July 2019. All three simulations included a 1-month spinup. An annotated map of the contiguous US is provided in Figure C1. The TROPOMI columns have high $NO_2$ concentrations over major cities and low $NO_2$ concentrations in rural and remote regions. Simulated $NO_2$ column densities from C180-global and C180e-US are consistent throughout the domain and generally
capture the plumes over cities and the low concentrations in rural and remote areas. Small differences, like those seen near Four Corners and Denver, can be attributed to differences from upscaling the emissions to the simulation grids (i.e., aliasing). In the

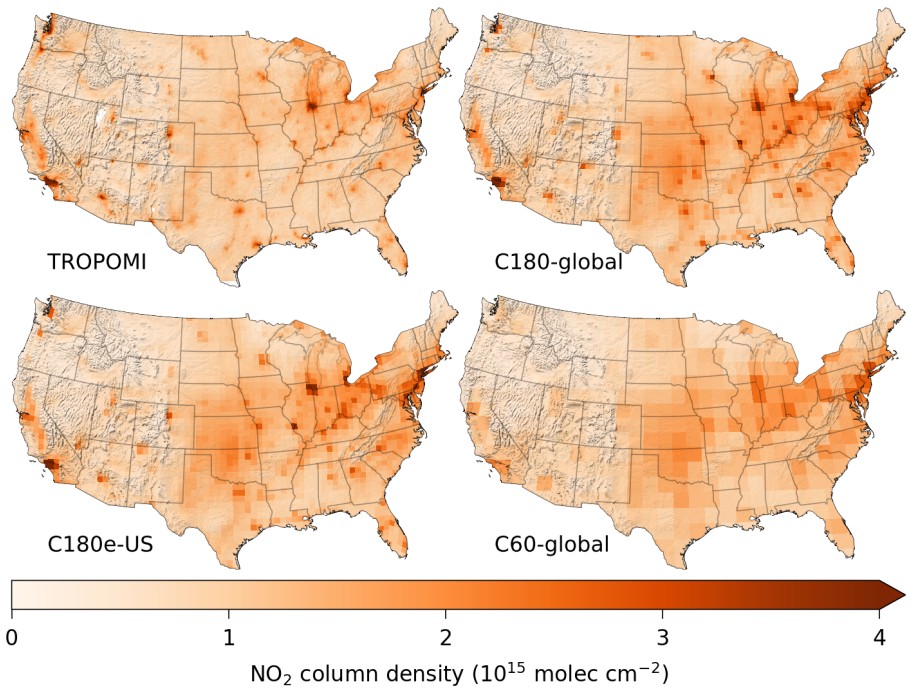

**Figure 7.** Mean tropospheric NO$_2$ column densities from C180-global, C180e-US, C60-global, and TROPOMI observations for July 2019. Simulated means only include points where TROPOMI observations were available. TROPOMI columns shown here use shape factors from C180e-US. TROPOMI columns with shape factors from C180-global were nearly identical. An annotated map of the contiguous US is provided in Figure C1.

case of the differences near Four Corners, emissions from natural gas production and power plants in the region have a spatial scale that is finer than the simulation grids. C60-global failed to resolve the local enhancements over major cities; comparison with C180e-US highlights the effectiveness of grid-stretching.

The computational demands of C180e-US were significantly less than C180-global and comparable to C60-global. Table 3 gives timing test results for C180-global, C180e-US, and C60-global. In terms of total computational workload, the total CPU time for C180e-US was ∼19 times less than C180-global. The improvement is better than one would approximate by simply considering the reduction in the total number of grid-boxes because the parallelization overhead is reduced as well. In terms of model throughput, C180e-US was 2.4 times faster than C180-global, despite using 8 times fewer cores. The total CPU time

for C180e-US was 15 % greater than C60-global. The slight increase of data input cost in C180e-US is suspected to be caused by a load imbalance in online input regridding in ESMF. Work to mitigate this load imbalance is underway.

## 3.2    A stretched-grid simulation with a large stretch-factor

A large stretch-factor creates a grid with a strong but localized refinement. Here, we experiment with using a large stretch-factor to simulate NO$_2$ columns in California at ∼10 km resolution. Simulated NO$_2$ columns in California are known to be

**Table 3.** Two-week timing test results comparing the computational expense of C180-global, C180e-US, and C60-global.

|  |  | C180-global | C180e-US | C60-global |
|---|---|---|---|---|
| Number of cores used | (CPUs) | 384 | 48 | 48 |
| Wall time |  |  |  |  |
| Chemistry | (hours) | 21.8 | 7.0 | 7.2 |
| Dynamics | (hours) | 3.1 | 1.7 | 1.7 |
| Data Input | (hours) | 0.5 | 1.8 | 0.3 |
| Other | (hours) | 0.3 | 0.2 | 0.2 |
| Total wall time | (hours) | 25.7 | 10.7 | 9.4 |
| Total CPU time | (days) | 411 | 21.3 | 18.7 |
| Model throughput | (days/day)[1] | 13.1 | 31.4 | 35.7 |

1. Simulation days per 24 wall hours.

sensitive to model resolution because of significant variability in emissions and topography (Valin et al., 2011). This case study is intended to probe the computational ability for very fine simulations, and the feasibility of using a highly stretched grid for an application that is well-suited for stretching.

For the stretched-grid simulation, we choose a grid size of C90 with stretching parameters $S = 10$, $T_\phi = 37.2$ °N, and $T_\theta = 119.5°$ W. We refer to this simulation as C900e-CA. The stretching parameters were chosen so that the target face approximately encompassed California. For comparison, we also conduct a cubed-sphere simulation with a C90 grid, which we refer to as C90-global. C90-global is effectively C900e-CA without grid-stretching. Both simulations use an identical model configuration, apart from the grid. Figure 8 compares the resolution of the two simulation grids. The average resolution of C900e-CA in California is 11.2 km. To understand the variability of the C900e-CA grid resolution it is useful to consider the local scaling at various locations. For example, New York is approximately 4000 km from C900e-CA's target point. Figure 3 shows that for $S = 10$, the local scaling at 4000 km is close to 1. Equivalently, substituting $S = 10$ and $\Theta = 4000$ km/$r_E$ in Eqn. 2 gives $L = 1.04$. Therefore, the resolution of C900e-CA in New York is similar to a standard C90 cubed-sphere. This can be confirmed in Figure 8. The combination of a large stretch-factor ($S = 10$) and moderate grid size (C90) causes some grid-boxes to become very coarse. Grid-boxes in the Indian Ocean and southeastern Africa are coarser than 1000 km. These coarse grid-boxes will result in an errors in $O_3$ production and an overestimation of stratosphere-troposphere exchange (Wild and Prather, 2006). Therefore, the C900e-CA grid would not be suitable for applications sensitive to $O_3$ or CO in the mid- and upper-troposphere. This underscores that selection of an appropriate stretched-grid is application-specific.

C900e-CA leverages the fine spatial resolution of anthropogenic NO emissions data available for the US; the NEI-2011 inventory has a 0.1°×0.1° grid (∼9 km). C900e-CA uses meteorological data from the GEOS-FP data product with a spatial resolution of 0.25°×0.3125° (∼25 km). We expect some of the detailed orographic and coastal effects in California to be missed. C900e-CA identifies a need for even finer resolution meteorological data, for which there is ongoing work in the

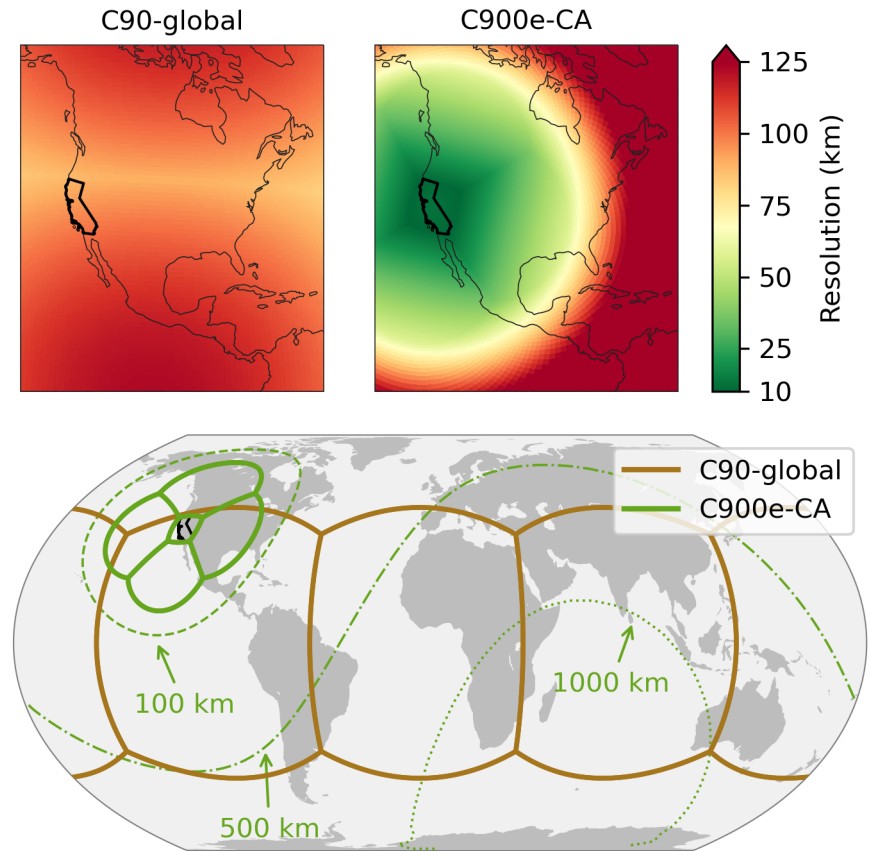

**Figure 8.** Comparison of C90-global and C900e-CA grids. The top panels show the variability of resolution for both grids. The bottom panel shows the grid face edges for C90-global and C900e-CA, with the 100 km, 500 km, and 1000 km resolution contours for C900e-CA.

GCHP–GMAO community. In the regions where C900e-CA's grid is finer than the GEOS-FP data, the data are conservatively downscaled (online by ESMF). We evaluate the effect of using donwscaled meteorological data on the simulated $NO_2$ columns in Appendix D and find no significant ill-effects for our application.

Figure 9 shows tropospheric $NO_2$ column densities from TROPOMI, C900e-CA, and C90-global for California in July 2019. Both simulations used a 1-month spinup. An annotated map of California is provided in Figure C2. The TROPOMI columns have significant fine-scale variability throughout California, high-concentrations over Los Angeles and in the San Francisco Bay Area, and smaller high-concentration features over Sacramento, Fresno, and Bakersfield. C900e-CA resolves many of the fine-scale spatial features seen in the TROPOMI columns, including the small high-concentration features over Sacramento, Fresno, and Bakersfield. The coarse resolution of C90-global fails to resolve most spatial features seen in the TROPOMI and C900e-CA columns, and significantly underestimates high concentrations except in Los Angeles (LA). The underestimation of high concentrations in C90-global, such as in the San Francisco Bay Area, is associated with the averaging of fine-scale urban emissions over the coarse grid-boxes. A subtle feature in the TROPOMI columns is the strong gradient

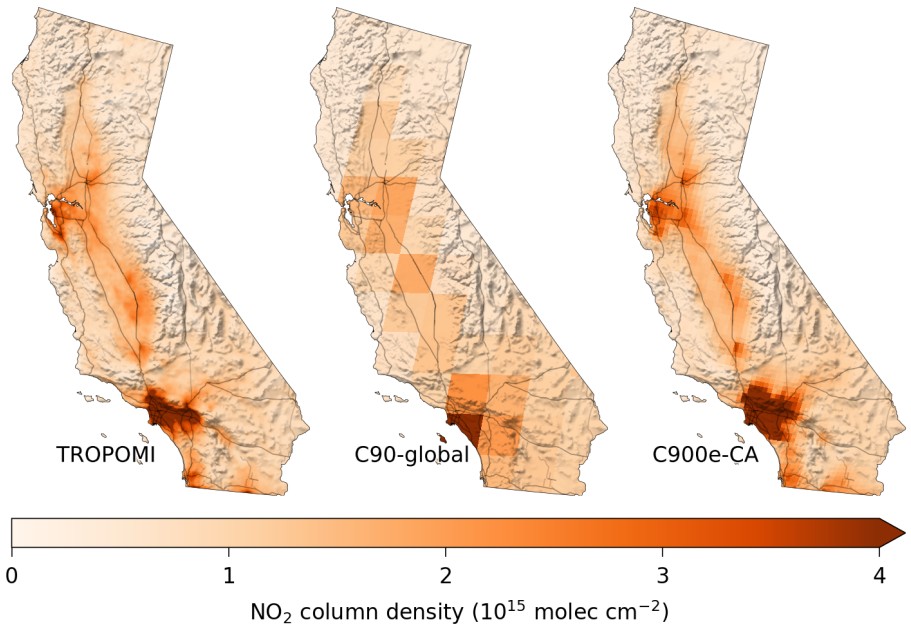

**Figure 9.** Mean tropospheric NO$_2$ column densities from C90-global, C900e-CA, and TROPOMI observations for July 2019. Simulated means only include points where TROPOMI observations were available. TROPOMI columns shown here use shape factors from C900e-CA. An annotated map of California is provided in Figure C2.

along the LA Basin perimeter. Neither simulation captures this gradient well, and LA's plume spuriously spreads into the Mojave Desert. The shallow inversion that prevents the plume from rising over the mountains might be better represented in finer resolution meteorological inputs that could better resolve coastal dynamics. Nonetheless, C900e-CA demonstrates a pronounced improvement over C90-global at resolving the challenging heterogeneity of California with similar computational expense (within 17 %).

## 4 Conclusions

Fine resolution simulations of atmospheric chemistry are necessary to capture fine-scale modes of variability such as localized sources, nonlinear chemistry, and boundary layer processes, but fine resolution simulations have been impeded by computational expense. This work developed stretched grids for the high-performance implementation of GEOS-Chem (GCHP). The capability was validated against global cubed-sphere simulations. Applications were probed with case studies that compared simulated concentrations with observations from the TROPOMI satellite instrument.

Stretched grids enable multiscale grids in GCHP while complimenting the other multiscale grid methods that are available in GEOS-Chem implementations. A primary benefit of grid-stretching is ease of use. The refinement is flexible and controlled by four simple runtime parameters. Stretched-grids operate naturally in GCHP, so switching between a stretched-grid and

cubed-sphere grid is seamless. Stretched-grid simulations are standalone simulations that do not require any pregenerated or dynamically-coupled boundary conditions. Compared to nested grids, the main disadvantages of grid-stretching are that there is a single refinement, and one cannot control the refined domain, refinement resolution, and global resolution independently.

Stretched-grid simulations can be used for regional- or continental-scale simulation purposes. Generally, stretch-factors in the range 1.4–4.0 are applicable for large refined domains. Higher stretch-factors can be used for very fine resolution simulations for regional-scale applications. To aid in choosing an appropriate stretch-factor, we propose a simple procedure based on choosing the maximum stretch-factor subject to two constraints: the size of the refined domain, and the maximum and minimum resolution.

Computationally, stretched-grid simulations are capable of unprecedented spatial resolutions for GEOS-Chem. Fine-scale emissions, with accurate spatial and temporal variability, and meteorological inputs with fine horizontal, temporal, and vertical resolution are needed to fully exploit these newly achievable resolutions. Stretched-grids are publicly available and ready for scientific application in GEOS-Chem version 13.0.0.

*Code availability.* GEOS-Chem is an open source project distributed under the MIT License at https://github.com/geoschem/GCHP. The
exact version of GCHP used in this manuscript was 13.0.0-alpha.3, and is archived on Zenodo (The International GEOS-Chem User Community, 2020).

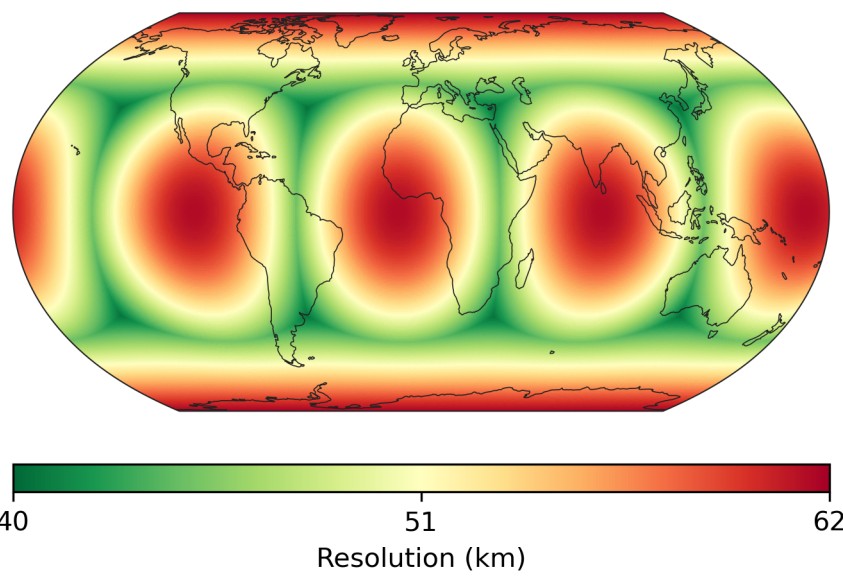

**Figure A1.** Variability of a C180 cubed-sphere's resolution.

### Appendix A: Variability of cubed-sphere grid resolution

Figure A1 shows the variability of a C180 cubed-sphere grid's resolution. The coarsest resolutions is at the center of faces, and the finest resolutions are at the corners of the faces. The average resolution of a cubed-sphere grid can be approximated by $\sqrt{A_E/(6 \times N \times N)}$, where $A_E$ is the surface area of Earth ($\sim 510 \times 10^6$ km$^2$) and $N$ is the size of the cubed-sphere (e.g., $N = 180$ for a C180 grid). The average resolution of a C180 grid is 51 km. The resolution at the center of the face is 62 km, and the resolution at the corner of the face is 41 km.

## Appendix B: Derivation of local scaling

Local scaling, $L$, is the relative change to a grid-box's length from grid-stretching. Consider a line segment that follows a meridian. The line segment starts at $y$ and has length $\Delta y$. After remapping the line segment with the Schmidt transform (Eqn. 1), the line segment starts at $y'$ and has length $\Delta y'$. The limit of the segment's local scaling as $\Delta y \to 0$ is equal to the derivative of the Schmidt transform:

$$L(y'; S) = \lim_{\Delta y \to 0} \frac{\Delta y'}{\Delta y} = \lim_{\Delta y \to 0} \frac{\phi'(y + \Delta y) - \phi'(y)}{\Delta y} = \frac{d}{dy} \phi'(y) \tag{B1}$$

The derivative of the Schmidt transform is

$$\frac{d}{dy} \phi'(y) = \frac{2S}{S^2(1 - \sin y) + \sin y + 1} \tag{B2}$$

so the local scaling at $y'$ is

$$L(y'; S) = \frac{2S}{S^2(1 - \sin y) + \sin y + 1} \tag{B3}$$

We can obtain the local scaling at $y$, rather than $y'$, by substituting the inverse of the Schmidt transform into Eqn. B3. This gives

$$L(y; S) = \frac{S^2(1 + \sin y) - \sin y + 1}{2S} \tag{B4}$$

Finally, we can generalize Eqn. B4 so it is a function of arclength from the target. The center of the refinement after applying Eqn. 1 is at the South Pole. The arclength from $y$ to the South Pole is $\Theta = y - (-\pi/2)$. Substituting $y = \Theta - \pi/2$ into Eqn. B4 gives

$$L(\Theta; S) = \frac{S^2(1 - \cos\Theta) + \cos\Theta + 1}{2S} \tag{B5}$$

## Appendix C: Maps of locations discussed in the text

Figure C1 shows a map of the contiguous US. Locations that are discussed in the text, and locations with notable features in Figure 7 are marked. Figure C2 shows a map of California. Locations that are discussed in the text, and locations with notable features in Figure 9 are marked.

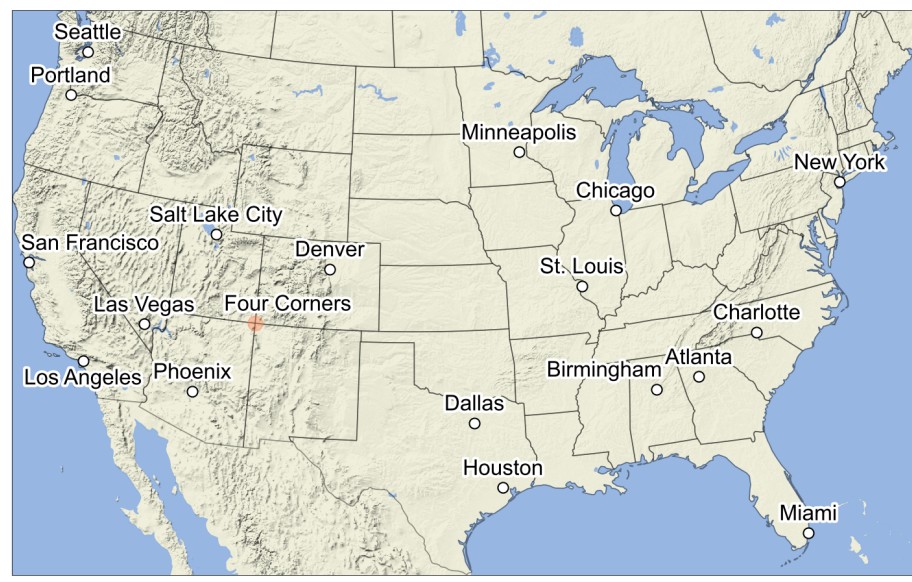

**Figure C1.** Map of the contiguous US. Locations discussed in the text and locations with notable features in Figure 7 are marked.

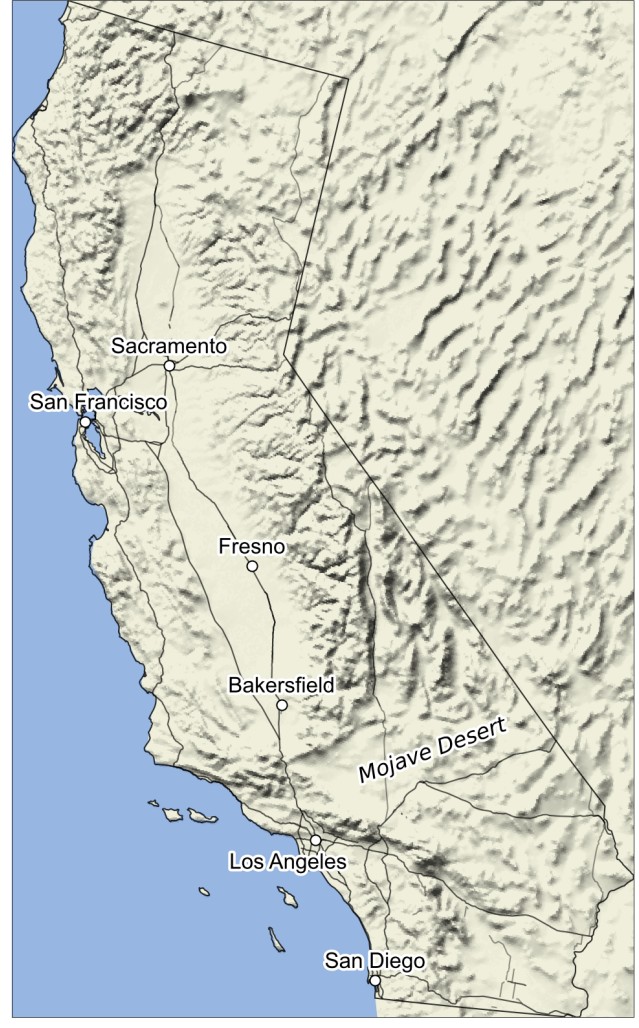

**Figure C2.** Map of the California. Locations discussed in the text and locations with notable features in Figure 9 are marked.

## Appendix D: Evaluation of interpolating meteorological inputs in California NO$_2$ simulation

Sect. 3.2 demonstrates the use of a stretched-grid simulation for simulating tropospheric NO$_2$ column densities in California. The simulation, C900e-CA, has an average resolution of 11.2 km in California, which is approximately twice as fine as the input meteorological data (0.25°×0.3125° grid, or ∼25 km resolution). To check that interpolating meteorological data in C900e-CA does not have significant unexpected consequences, we downscaled the data to a 0.5°×0.625° grid (∼50 km) and conducted a pair of stretched-grid simulations with ∼25 km resolution; one used the 0.25°×0.3125° data and the other used

the downscaled 0.5°×0.625° data. The NO$_2$ columns from these simulations are shown in Figure D1 (second row), along with the columns from TROPOMI and C900e-CA (first row). The grid for these simulations was a C60 cubed-sphere with a stretch-factor of $S = 6$ (∼25 km) and a target point of 37.2 °N, 119.5 °W (same as C900e-CA). Both used a 1-month spinup. Comparison of the lower panels in Figure D1 shows that interpolating the 0.5°×0.625° meteorological inputs to the ∼25 km grid has little effect on the simulated NO$_2$ columns. This suggests the consequences of interpolating meteorological

data in C900e-CA are minor for our application, and that the variability is driven by the resolution of emissions data (∼9 km resolution).

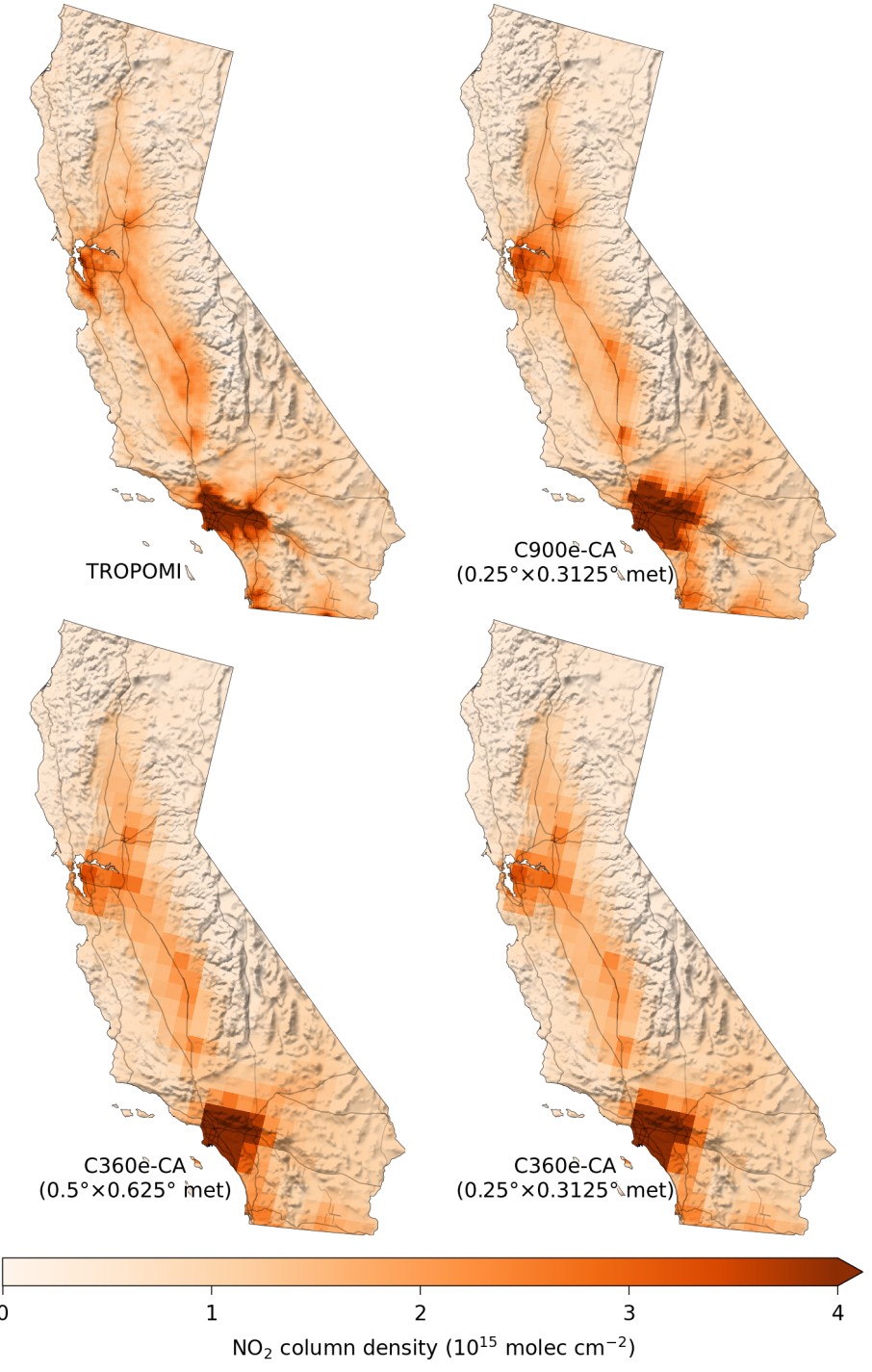

**Figure D1.** Mean tropospheric NO₂ column densities from TROPOMI, C900e-CA, C360e-CA with $0.25° \times 0.3125°$ meteorological data, and C360e-CA with meteorological data downscaled to $0.5° \times 0.625°$.

*Author contributions.* LB implemented and validated the stretched-grid capability in GCHP. RVM, DJJ, and SP provided project oversight and top-level design. MJC reprocessed TROPOMI columns with shape factors from simulations. EWL, LB, and SDE performed MAPL 2 upgrade in GCHP. BMA and TLC developed stretched-grid capability in MAPL 2. WMP developed FV3. HW, JL, LTM, and JM developed grid-independent emissions for GEOS-Chem, and CAK developed HEMCO. LB wrote the manuscript. All authors contributed to manuscript editing and revisions.

*Competing interests.* The authors declare that they have no conflict of interest.

*Acknowledgements.* This work was supported by a NASA AIST program grant (18-AIST18-0011), and by the Natural Sciences and Engineering Research Council of Canada (RGPIN-2019-04670).

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
