# Peer review of "Grid-Stretching Capability for the GEOS-Chem 13.0.0 Atmospheric Chemistry Model"

_Geoscientific Model Development, 2020_

## Referee Comment (RC1) · Anonymous Referee #1 · 22 Feb 2021

General Comments

The manuscript describes the introduction of stretched-grids for GEOS-Chem in its high-performance implementation GCHP. It demonstrates its abilities in high-resolution modeling by performing two refined simulations, one for the contiguous US using broad refinement, and one for California with a strong refinement of about 10 km. Comparison with TROPOMI tropospheric NO2 retrievals reveals reasonable spatial agreement of the simulated columns, while computational expenses can be cut down significantly in comparison with a global high-resolution run.

Overall, I read the manuscript with curiosity. Grid-stretching, when implemented in global chemistry models benefiting from a HPC environment, has the potential to overcome the barriers which practically limit the models' resolution due to missing computational resources and model parallelism. Nested simulations are the classical approach, but are hampered by the necessity to define and transfer adequate initial and boundary conditions or coupling parameters between the model domains. Grid-stretching is seamless and easy to use, once the capabilities have been set up properly. The authors' work builds up on a series of recent developments for GEOS-Chem and its components, including grid-independent chemical transport, introduction of stretched-grids for the FV3 dynamical core, improved parallelism and coupling of model components, and the use of grid-independent emission inventories. Here, for the first time all these developments have been combined to allow for high-resolution refinement of chemistry simulations, technically feasible for localized resolutions down to 10 km. Although more validation of the model setup will be needed, e.g. by comparison with in situ measurements, by using equally high resolution meteorological data, or by evaluating transport and diffusion of longer-lived species, I recommend to publish the paper without such extensions but after responding to some minor corrections which are outlined below.

Specific Comments

Introduction

Please give some more overview on other grid refinement techniques, like adaptive mesh refinement (e.g, Slingo et al., 2009; Garcia-Menendez and Odman, 2011) and parent-child coupling (Zängl et al., 2015).

Lines 17-19: Please add production and destruction by chemical reaction.

Lines 19-20: In addition to missing computational resources, models typically also lack sufficient model parallelism, making high resolution simulations inefficient or even unfeasible.

Lines 26-29: Add a few more references which reflect pioneer or developments apart from GEOS-Chem, e.g. Miyakoda and Rosati (1977), Zhang et al. (1986), Krol et al. (2005).

Development of Stretched-Grids in GEOS-Chem

For better orientation to users not familiar with GEOS-Chem: Please add an overview of used cubed-sphere sizes together with an estimate of model resolution in km. This can be given as a table or as a formula if apllicable.

Line 60: Refences should be given already in the introduction.

Lines 64-65: Add references to Rienecker et al. (2008), Todling and El Akkraoui (2018).

Lines 81-82: Re-formulate this sentence. A rotation in latitude (longitude) direction must result in a new target latitude (longitude).

Lines 115-116: Re-formulate this sentence: "A constraint for S, such that the refined domain diameter is greater than wtf . . ."

Line 126: While the constraints given in this section are clear to me, I cannot follow the choice of C104 here. How is this number derived?

Lines 128-131: All emissions presented here are for the US only. What about emissions outside this region? They may play a role for transported species.

Lines 130-131: Add reference to van der Werf et al. (2017).

Lines 145-149: This rather short section raises expectations, that either you will explain next the prototype simulation or the benchmarking code, but you don't follow this path further. Either this section needs to be extended by some more details or even more shortened.

Lines 154-156: If I understood correctly, you chose a stretch factor of 2.4 to achieve a similar resolution of C96-global and C96e-NA over the US. The target point is chosen so that the grid boxes have minimal overlap over the US. Please state this more clearly.

Lines 162-166: Have all simulations been performed using exactly the same time step, vertical resolution, meteorological and chemical input data?

Stretched-Grid Case-Studies

Lines 171-173: I would not call this an assimilation of meteorological data into GEOS-Chem, as the text suggests, it's rather a kind of nudging.

Lines 198-201: For readers not familiar with US geography: Please specify location and type of source for emissions at "Four Corners" and "near Denver". I argue these emissions are from large power plants. Both locations could also be marked in Fig. 6.

Lines 210-211: Can you give a reference?

Lines 211-214: It should be noted here that a resolution of 9-10 km is at the lower edge of what can be simulated using traditional model physics parameterizations.

Lines 216-217: It would indeed be interesting to compare your simulations with a run using meteorological input with higher horizontal, vertical, and temporal resolution.

Line 224: Is any spinup used for this simulation?

Lines 225-228: Please mark less well known cities in Figure 8.

Conclusions

Lines 241-242: Which grid methods exactly can be used by GEOS-Chem?

Appendix A

Line 261: Is it possible to give a formula for deriving the average resolution?

Figures

Figure 1: You could add for illustration, that the number of grid faces is always 6.

Figure 3: "C94-global" can be omitted in the lower panel.

Figure 4: Chemical species are given in mixing ratios, not as concentrations. Please specify mole mole-1 or kg kg-1.

Figure 6: Please mark locations of Denver and Four Corners. What can be said about the point source in western Wyoming, which is only visible in the high resolution simulations?

Figure 8: Please mark locations of Sacramento, Fresno, and Bakersfield. Why are emissions in the San Francisco Bay area largely underestimated in C90-global?

Technical Corrections

Lines 277-278: Exchange "B5" by "B4" (two times).

References:

Garcia-Menendez, F. and Odman, M. T.: Adaptive grid use in airquality modeling, Atmosphere, 2, 484–509, 2011.

Krol, M., Houweling, S., Bregman, B., van den Broek, M., Segers, A., van Velthoven, P., Peters, W., Dentener, F., and Bergamaschi, P.: The two-way nested global chemistry-transport zoom model TM5: algorithm and applications, Atmos. Chem. Phys., 5, 417–432, https://doi.org/10.5194/acp-5-417-2005, 2005.

Miyakoda, K., and Rosati, A.: One-way nested grid models: The interface conditions and the numerical accuracy, Mon. Weather Rev., 105, 1092-1107, 1977.

Rienecker, M. M., and Coauthors, 2008. The GEOS-5 data assimilation system—-Documentation of versions 5.0.1, 5.1.0, and 5.2.0. NASA/TM-2008-104606, Vol. 27, 118 pp., https://gmao.gsfc.nasa.gov/pubs/docs/GEOS5_104606-Vol27.pdf.

Todling, R., and A. El Akkraoui, 2018: The GMAO hybrid ensemble-variational atmospheric data assimilation system: Version 2.0. NASA/TM-2018-104606, Vol. 50, 184 pp., https://gmao.gsfc.nasa.gov/pubs/docs/Todling1019.pdf.

van der Werf, G. R., Randerson, J. T., Giglio, L., van Leeuwen, T. T., Chen, Y., Rogers, B. M., Mu, M., van Marle, M. J. E., Morton, D. C., Collatz, G. J., Yokelson, R. J., and Kasibhatla, P. S.: Global fire emissions estimates during 1997–2016, Earth Syst. Sci.

[Figure]

Data, 9, 697–720, https://doi.org/10.5194/essd-9-697-2017, 2017.

Zängl, G., Reinert, D., Rípodas, P., and Baldauf, M.: The ICON (ICOsahedral Non-hydrostatic) modelling framework of DWD and MPI-M: Description of the non-hydrostatic dynamical core, Q. J. Roy. Meteor. Soc., 141, 563–579, doi:10.1002/qj.2378,2015.

Zhang, D. L., Chang, H. R., Seaman, N. L., Warner, T. T., and Fritsch, J. M.: A two-way interactive nesting procedure with variable terrain resolution, Mon. Weather Rev., 114, 1330–1339, 1986.

---

## Referee Comment (RC2) · Anonymous Referee #2 · 24 Feb 2021

General Comments

The manuscript describes the advantage of stretch-type grid refinement in atmospheric chemical transport models. The simulation and computational performances of a stretched-grid experiment are evaluated compared to those of a global quasi-uniform grid with similar horizontal resolution. The effectiveness of the more aggressive grid refinement in California with strong pollutant emissions and complex topography was also evaluated by comparing TROPOMI tropospheric NO2 column densities. The methodology is straightforward and the results are concise. However, the manuscript's content looks like a technical note on using the GEOS-Chen model in stretch mode. I could find neither substantial advances in chemical transport modeling nor novel ideas to share with the reader. In my opinion, it requires major revision before ready for publication.

[Figure]

As the authors mentioned in the introduction, one of the advantages of using stretch grids is that down-scaling similar to 2-way nesting can be easily achieved without lateral boundaries. However, there are also side effects when reducing the spatial resolution outside the target region: In Figure 4, there appears to be a systematic bias in the ozone concentration in the free troposphere. It is necessary to investigate whether such a bias is due to changes in ozone precursors' transport pathways outside the focal region. With the recent improvement in computer performance, atmospheric simulations using thousands or tens of thousands of CPU cores are no longer unusual. In terms of time-to-solution, the stretch calculation of C180e-US is only 2.5 times faster than that of C180-global. The authors should consider whether they will tolerate the side effects and reduce computing resource consumption by just a few hundred cores.

In more aggressive stretching experiments, the advantage of high resolution should be examined from multiple perspectives. Many regional chemical transport simulations have already shown that higher resolution emission data can reproduce the tracer concentration distribution near the emission source. The authors should emphasize the advantages of the grid-stretched global chemical transport model that are not present in the regional model. For example, I want to know what advantage there is in keeping the resolution around New York at the same degree as in the non-stretch experiment. Besides, the author should evaluate that they chose a higher resolution than the meteorological field. I recommend adding a stretch calculation experiment with a resolution close to that of the meteorological field. If that resolution gives almost the same results as the 10km resolution in this study's evaluation method, there is no need to use the 10km resolution.

Specific Comments

Introduction: Although it is mentioned in Harris et al.(2016), Tomita(2008) should be referred to, as a previous study of global atmospheric models with stretch grids using the Shumidt transform. The studies of Goto et al.(2015) and Trieu et al.(2017), which simulated atmospheric chemical transport on the stretched grid, should also be cited.

[Figure]

Lines 38-46: It seems to me that almost all the tools to realize stretch calculations were available before this study, and there are no technical problems to be solved to perform chemical transport calculations on a stretched-grid. I would like to see a clarification of what the authors contributed to the implementation.

Lines 65-66: Is the meteorological field re-gridded for both horizontal and vertical, and if so, how is the transformation applied between model grids with different topography?

Lines 66-67: Does the model solve the stratospheric chemical processes, and if not, how is the stratospheric gas (ozone) concentration given?

Lines 135-136: The number of vertical layers and each layer's altitude in the meteorological field is the same as in the GEOS-Chem settings? How much higher is the top altitude for the meteorological field than for the GEOS-Chem simulations?

Lines 136-137: For which time period was this simulation performed?

Lines 154-155: I want to know why the authors did not align the Face center of the hexahedron between the focal region of the C96e-NA and the C96-global.

Lines 163-164, Figure4: 1) In order to clarify the difference in the free troposphere, please show the results for the layer near the surface and the layer above the surface separately. 2) As mentioned in the major comment, the difference in ozone concentration in the free troposphere (left panel) appears to be bias rather than scatter. I would like to see further analysis of this. 3) In the figure for OH, the colors are largely different between the right and left figures. In the left figure, where did the red dot around 0.1-0.2ppt go?

Lines 204-205, Table2: 1) The differences in the number of grid points and the number of time steps (time intervals) between the C180-global and C60-global experiments are not described. Since the relative difference in computational workload can be approximated by the number of grid points and the number of time steps, this information is necessary to evaluate the computational performance. 2) It is not stated how many

months of calculation results were used for the comparison. Recently, the unit "Simulation Year Per wall clock Day (SYPD)" is often used to express the elapsed time of a simulation. This should be added.

Lines 224-225, Figure8: Some people don't even know which is north for LA or SF. I don't know which area is the Mojave Desert. Please add a mark to the figure.

Lines 231-233: 1) I interpreted that the authors upscaled the 0.25degx0.3125deg meteorological data to a grid resolution of ~100 km in the C90-global experiment and interpolated it to a grid resolution of ~10 km in the C900e-CA. Does this mean that a spatial resolution of 0.25degx0.3125deg is sufficient to represent the concentration gradient around LA? 2) To discuss why the concentration gradients were well reproduced, it is necessary to evaluate whether the higher resolution of the emission or the higher resolution of the meteorological field was more effective. Previous researches should be presented. Or sensitivity experiments should be conducted, such as using upscaled, low-resolution emission inventories or meteorological fields in high-resolution experiments. 3) GEOS-Chem is a transport model, and the wind field is a given. I want the authors to show the benefits of usage of higher resolution than the meteorological field for vertical transport, diffusion and chemical reaction in detail. The 10km-mesh simulation provide a better representation of the concentration gradient near the emission source than the 0.25deg-mesh? I would like to see an analysis of the ozone production/dissipation balance, etc.

References

H. Tomita (2008), A stretched grid on a sphere by new grid transformation, J. Meteorol. Soc. Jpn., 86A, 2008, pp. 107-119

D. Goto, T. Dai, M. Satoh, H. Tomita, J. Uchida, S. Misawa, T. Inoue, H. Tsuruta, K. Ueda, C.F.S. Ng, A. Takami, N. Sugimoto, A. Shimizu, T. Ohara, T. Nakajima (2015), Application of a global nonhydrostatic model with a stretched-grid system to regional aerosol simulations around Japan, Geosci. Model Dev., 8, 2015, pp. 235-259,

doi:10.5194/gmd-8-235-2015

T.T.N. Trieu, D. Goto, H. Yashiro, R. Murata, K. Sudo, h. Tomita, M. Satoh, T. Naka-jima (2017), Evaluation of summertime surface ozone in Kanto area of Japan using a semi-regional model and observation, Atmos. Environ., 153, 2017, pp. 163-181, https://doi.org/10.1016/j.atmosenv.2017.01.030

---

## Author Comment (AC1) · 26 Jun 2021

We have uploaded our responses to RC1 and RC2 in the form of a supplement. Please see the attached document below.

Please also note the supplement to this comment: https://gmd.copernicus.org/preprints/gmd-2020-398/gmd-2020-398-AC1-supplement.pdf

---

## Author Response (AR1)

**Author response to referee comments (gmd-2020-398)**

We would like to thank both anonymous referees for their time and thoughtful comments. Below we include point-by-point responses to each referee comment in blue type and proposed revisions to the manuscript in *blue italic type*.

We have carefully revised the manuscript to:
1. improve the clarity of its scope, novelty, and logic;
2. add a review of other refinement techniques and pioneering works;
3. expand the model description, including an overview of the model's standard horizontal grid and input data.

We believe the responses and proposed revisions, detailed below, address all of the referees' comments.

**Anonymous Referee #1 (Rev. 1)**

**General comments**

[Rev. 1, preamble]  The manuscript describes the introduction of stretched-grids for GEOS-Chem in its high-performance implementation GCHP. It demonstrates its abilities in high-resolution modeling by performing two refined simulations, one for the contiguous US using broad refinement, and one for California with a strong refinement of about 10 km. Comparison with TROPOMI tropospheric NO2 retrievals reveals reasonable spatial agreement of the simulated columns, while computational expenses can be cut down significantly in comparison with a global high-resolution run.

Overall, I read the manuscript with curiosity. Grid-stretching, when implemented in global chemistry models benefiting from a HPC environment, has the potential to overcome the barriers which practically limit the models' resolution due to missing computational resources and model parallelism. Nested simulations are the classical approach, but are hampered by the necessity to define and transfer adequate initial and boundary conditions or coupling parameters between the model domains. Grid-stretching is seamless and easy to use, once the capabilities have been set up properly. The authors' work builds up on a series of recent developments for GEOS-Chem and its components, including grid-independent chemical transport, introduction of stretched-grids for the FV3 dynamical core, improved parallelism and coupling of model components, and the use of grid-independent emission inventories. Here, for the first time all these developments have been combined to allow for high-resolution refinement of chemistry simulations, technically feasible for localized resolutions down to 10 km. Although more validation of the model setup will be needed, e.g. by comparison with in situ measurements, by using equally high resolution meteorological data, or by evaluating transport and diffusion of longer-lived species, I recommend to publish the paper without such extensions but after responding to some minor corrections which are outlined below.

> Thank you for this insightful perspective.

**Specific comments**

**Section 1: Introduction**

[Rev. 1, comment 1.] Please give some more overview on other grid refinement techniques, like adaptive mesh refinement (e.g, Slingo et al., 2009; Garcia-Menendez and Odman, 2011) and parent-child coupling (Zängl et al., 2015).

> We have added a review of refinement techniques to the introduction (paragraph 2). We include Zängl et al., 2015 under the umbrella of grid-nesting, since it could be considered a type of two-way nesting. We thank the reviewer for suggesting an expanded overview of refinement techniques.

[Rev. 1, comment 2.] Lines 17-19: Please add production and destruction by chemical reaction.

> We have added the suggested phrasing.

**[Rev. 1, comment 3.]** Lines 19-20: In addition to missing computational resources, models typically also lack sufficient model parallelism, making high resolution simulations inefficient or even unfeasible.

> Good point. We rephrased the sentence to reflect that in general it is the degree of parallelism in a model that limits its resolution.
>
> > *Typical global model resolutions are on the order of hundreds of kilometers and generally limited by the degree of model parallelism.*

**[Rev. 1, comment 4.]** Lines 26-29: Add a few more references which reflect pioneer or developments apart from GEOS-Chem, e.g. Miyakoda and Rosati (1977), Zhang et al. (1986), Krol et al. (2005).

> We have added a review of refinement techniques to the introduction (paragraph 2), and we include references to these important works.

**Section 2: Development of Stretched-Grids in GEOS-Chem**

**[Rev. 1, comment 5.]** For better orientation to users not familiar with GEOS-Chem: Please add an overview of used cubed-sphere sizes together with an estimate of model resolution in km. This can be given as a table or as a formula if applicable.

> We added a table to Sect. 2.1 that compares common GCHP grids and regular lat-lon grids. In Sect. 2.1 we also expanded the description of the cubed-sphere grid.

**[Rev. 1, comment 6.]** Line 60: References should be given already in the introduction.

> We have added references to Suarez et al. (2007), and Hill et al. (2004) to the initial mentions of MAPL and ESMF in the introduction.

**[Rev. 1, comment 7.]** Lines 64-65: Add references to Rienecker et al. (2008), Todling and El Akkraoui (2018).

> Done.

**[Rev. 1, comment 8.]** Lines 81-82: Re-formulate this sentence. A rotation in latitude (longitude) direction must result in a new target latitude (longitude).

> We thank the reviewer for pointing out the troublesome notation and phrasing of this description. We have changed the notation to $T_\phi$ and $T_\theta$ for the target latitude and target longitude, to emphasize that they are parameters of the stretching operation, and to distinguish them from the variables used in Eqn. (1).

**[Rev. 1, comment 9.]** Lines 115-116: Re-formulate this sentence: "A constraint for S, such that the refined domain diameter is greater than wtf . . ."

> We have rephrased the description of this constraint:
>
> > *For a target-face with width greater than $w_{tf}$ , S must satisfy $S \leq 0.414\, cot\, (w_{tf}\, /\, 4\, r_E)$ where $r_E$ is Earth's radius.*

[Rev. 1, comment 10.] Line 126: While the constraints given in this section are clear to me, I cannot follow the choice of C104 here. How is this number derived?

We have added an explanation of how C104 was chosen,

*To determine the grid size and stretch-factor for a simulation from these constraints, first, assume S = 3.5 as an initial value. For a maximum resolution comparable to a C360 grid with S = 3.5, the grid size would be C102.9 (360/3.5 = 102.9). Since the grid size must be an even integer, one should round up to C104 and choose S = 3.46 (360/104 = 3.46).*

[Rev. 1, comment 11.] Lines 128-131: All emissions presented here are for the US only. What about emissions outside this region? They may play a role for transported species.

We have added the inventories representing Asian and global $NO_x$ emissions (along with other anthropogenic emissions). We also clarified that this table details $NO_x$ sources. This table focuses on $NO_x$ emissions because of their strong influence on the results in Sect 3.1 and 3.2.

[Rev. 1, comment 12.] Lines 130-131: Add reference to van der Werf et al. (2017).

We have added the reference to the emission source summary table (Table 2), and to the description of the emissions used in the common model configuration.

[Rev. 1, comment 13.] Lines 145-149: This rather short section raises expectations, that either you will explain next the prototype simulation or the benchmarking code, but you don't follow this path further. Either this section needs to be extended by some more details or even more shortened.

We thank the reviewer for this feedback. We decided to remove this paragraph and focus Sect. 2.5 solely on the stretched-grid versus cubed-sphere comparison of oxidants and PM2.5 in the contiguous US. As a result, we also rename Sect. 2.5 to "Validating the stretched-grid capability".

[Rev. 1, comment 14.] Lines 154-156: If I understood correctly, you chose a stretch factor of 2.4 to achieve a similar resolution of C96-global and C96e-NA over the US. The target point is chosen so that the grid boxes have minimal overlap over the US. Please state this more clearly.

We thank the reviewer for pointing out that our reason for choosing a stretch-factor of 2.4 was not clear. The stretch-factor (S=2.4) was chosen so the average resolution of C96e-NA in the US was the same as the average resolution of C96-global in the US. The fact that the grid-boxes have little overlap is a consequence of choosing two grids with a similar resolution. We have clarified the description of our reasoning for S=2.4 in the text.

*The stretched-grid simulation, C96e-NA, has a grid size of C48 and stretching parameters S = 2.4, $T_\phi$ = 35° N, and $T_\theta$ = 96° W. The target point was chosen so the target-face approximately encompassed the populous regions of North America. The stretch-factor was chosen so that the average resolution of C96e-NA was equal to the average resolution of C96-global in the contiguous*

*US; we note that the stretch-factor is 2.4, rather than 2.0, because the US is a region where the standard cubed-sphere grid has a finer resolution than its global average as shown in Appendix A.*

[Rev. 1, comment 15.] Lines 162-166: Have all simulations been performed using exactly the same time step, vertical resolution, meteorological and chemical input data?

Yes, that is correct. All the simulations in the manuscript use the same model configuration (except for the grid). To avoid confusion, we updated the model configuration description (Sect. 2.4) so that it states this more clearly:

*The simulations in this manuscript use a shared model configuration of GCHP version 13.0.0-alpha.3. … All simulations use a 10-minute timestep for chemistry and a 5-minute timestep for transport.*

We also added remarks to reiterate that the simulations use a common model configuration Sect. 2.5 (validation simulations),

*All three simulations used an identical model configuration, apart from the grid, which is described in Sect. 2.4.*

to Sect. 3.1 (simulations for US case study),

*Both simulations use an identical model configuration, apart from the grid.*

and to Sect. 3.2 (simulations for California case study):

*Both simulations use an identical model configuration, apart from the grid.*

**Section 3: Stretched-Grid Case-Studies**

[Rev. 1, comment 16.] Lines 171-173: I would not call this an assimilation of meteorological data into GEOS Chem, as the text suggests, it's rather a kind of nudging.

Good point. This phrase was removed in a separate update to make the purpose of case studies more clear. However, we note and appreciate the correction.

[Rev. 1, comment 17.] Lines 198-201: For readers not familiar with US geography: Please specify location and type of source for emissions at "Four Corners" and "near Denver". I argue these emissions are from large power plants. Both locations could also be marked in Fig. 6.

We have added an annotated map in Appendix C (Figure C1). We confirmed that the emissions near Four Corners are from power plants and natural gas emissions. We added a remark to this section stating this.

*Small differences, like those seen near Four Corners and Denver, can be attributed to differences from upscaling the emissions to the simulation grids (i.e., aliasing). In the case of the differences near Four Corners, emissions from natural gas production and power plants in the region have a spatial scale that is finer than the simulation grids.*

[Rev. 1, comment 18.] Lines 210-211: Can you give a reference?

We have added a reference to Valin et al., (2011).

[Rev. 1, comment 19.] Lines 211-214: It should be noted here that a resolution of 9-10 km is at the lower edge of what can be simulated using traditional model physics parameterizations.

We thank the reviewer for the suggestion, but respectfully, we do not think this remark is necessary. Freitas et al., (2018) and Freitas et al., (2020) have developed and evaluated parameterizations for GEOS that are suitable to scales finer than 10 km (at least as fine as 3 km).

References

Freitas, S. R., Grell, G. A., Molod, A., Thompson, M. A., Putman, W. M., Santos e Silva, C. M., & Souza, E. P. (2018). Assessing the Grell-Freitas convection parameterization in the NASA GEOS modeling system. *Journal of Advances in Modeling Earth Systems*, 10, 1266– 1289. https://doi.org/10.1029/2017MS001251

Freitas, S. R., Putman, W. M., Arnold, N. P., Adams, D. K., & Grell, G. A. (2020). Cascading toward a kilometer-scale GCM: Impacts of a scale-aware convection parameterization in the Goddard Earth Observing System GCM. *Geophysical Research Letters*, 47, e2020GL087682. https://doi.org/10.1029/2020GL087682

[Rev. 1, comment 20.] Lines 216-217: It would indeed be interesting to compare your simulations with a run using meteorological input with higher horizontal, vertical, and temporal resolution.

Indeed, this would be interesting and will be explored in future work.

[Rev. 1, comment 21.] Line 224: Is any spinup used for this simulation?

C900e-CA and C90-global used a 1-month spinup. We clarified this in the text.

[Rev. 1, comment 22.] Lines 225-228: Please mark less well known cities in Figure 8.

We added an annotated map of California in Appendix C (Figure C2).

**Section 4: Conclusions**

[Rev. 1, comment 23.] Lines 241-242: Which grid methods exactly can be used by GEOS-Chem?

We have added a remark in the introduction that states that GEOS-Chem Classic and WRF-GC both support one-way and two-way nesting.

*One-way and two-way nesting are supported by the single-node version of GEOS-Chem "Classic" (Wang et al., 2004; Yan et al., 2014), and by GEOS-Chem using Weather Research and Forecasting (WRF) meteorology (WRF-GC; Lin et al. (2020); Feng et al. (2020)).*

**Appendix A**

[Rev. 1, comment 24.] Line 261: Is it possible to give a formula for deriving the average resolution?

> The average resolution can be approximated as $\sqrt{A_E / (6 \times N \times N)}$ where $N$ is the cubed-sphere size and $A_E$ is the surface area of Earth. We have added this to Appendix A.

**Appendix B**

[Rev. 1, comment 25.] Lines 277-278: Exchange "B5" by "B4" (two times).

> Corrected. Thanks.

**Figures**

[Rev. 1, comment 26.] Figure 1: You could add for illustration, that the number of grid faces is always 6.

> In response to Rev. 1, comment 5 we added a figure (now Figure 1) to supplement our expanded description of the GCHP grid. In this new figure's caption we added a remark that cubed-sphere grids always have six faces.

[Rev. 1, comment 27.] Figure 3: "C94-global" can be omitted in the lower panel.

> Done.

[Rev. 1, comment 28.] Figure 4: Chemical species are given in mixing ratios, not as concentrations. Please specify mole mole-1 or kg kg-1.

> Corrected to mol mol-1.

[Rev. 1, comment 29.] Figure 6: Please mark locations of Denver and Four Corners. What can be said about the point source in western Wyoming, which is only visible in the high resolution simulations?

> We thank the reviewer for noting the point source in western Wyoming. We found that there was an error in 13.0.0-alpha.3 that caused GFED4 emissions for 2016 to be used in the simulations (rather than the emissions for 2018).

> After fixing this error, we also decided to change the simulations in Sect. 3.1 and 3.2 to June--July 2019 (rather than 2018), due to uncertainties in emissions and observations associated with the intense wildfires in California in 2018.

> We corrected the error, reran the simulations, and we have updated the figures. We have also added markings for Denver and Four Corners to Figure C1.

[Rev. 1, comment 30.] Figure 8: Please mark locations of Sacramento, Fresno, and Bakersfield.

> We have marked these locations in Figure C2.

[Rev. 1, comment 31.] … Why are emissions in the San Francisco Bay area largely underestimated in C90-global?

The underestimation of NO2 concentrations in the San Francisco Bay Area is associated with grid-averaging (the emissions are averaged across the coarse grid-boxes that cover the area). We have added a remark that states this.

*The underestimation of high concentrations in C90-global, such as in the San Francisco Bay Area, is associated with the averaging of fine-scale urban emissions over the coarse grid-boxes.*

**Anonymous Referee #2 (Rev. 2)**

[Rev. 2, preamble] The manuscript describes the advantage of stretch-type grid refinement in atmospheric chemical transport models. The simulation and computational performances of a stretched-grid experiment are evaluated compared to those of a global quasi-uniform grid with similar horizontal resolution. The effectiveness of the more aggressive grid refinement in California with strong pollutant emissions and complex topography was also evaluated by comparing TROPOMI tropospheric NO2 column densities.

[Rev. 2, comment 1] The methodology is straightforward and the results are concise. However, the manuscript's content looks like a technical note on using the GEOS-Chen model in stretch mode. I could find neither substantial advances in chemical transport modeling nor novel ideas to share with the reader. In my opinion, it requires major revision before ready for publication.

> We thank the reviewer for their time and thoughtful feedback. We have incorporated suggestions that were made by the reviewer. We believe these changes not only improve the quality of the manuscript, but contribute to the clarity about its novelty.

> Our manuscript is a model improvement type manuscript in which we develop and describe a grid refinement capability for GEOS-Chem High Performance using grid stretching. We now clarify that this is one of the first manuscripts to apply grid stretching to an offline CTM, and the first to apply it to GEOS-Chem.

[Rev. 2, comment 2.a.] As the authors mentioned in the introduction, one of the advantages of using stretch grids is that down-scaling similar to 2-way nesting can be easily achieved without lateral boundaries. However, there are also side effects when reducing the spatial resolution outside the target region: In Figure 4, there appears to be a systematic bias in the ozone concentration in the free troposphere. It is necessary to investigate whether such a bias is due to changes in ozone precursors' transport pathways outside the focal region.

> We thank the reviewer for this comment and agree that the effects of the coarse grid outside the refinement are an important consideration. Constraint 2 in Sect. 2.3 is specifically included for this reason. There is a potential for resolution-dependent biases in CO and $O_3$ to be introduced by the coarse grid outside the refined domain; sensitivity to such effects is application-specific, so a means of constraining $S$ according to a "coarsest allowable resolution" is necessary ($R_{min}$). Applications that are sensitive to such biases, e.g. a simulation of $O_3$ aloft, should choose less aggressive stretching parameters (by a fine $R_{min}$ constraint), or consider a standard cubed-sphere.

> Figure 4 is a test to validate the technical implementation, so a moderate stretch-factor is desirable so that the stretched-grid capability is well-exercised in the test. The small low bias is consistent with a general tendency of coarse simulations to underestimate $O_3$ concentrations aloft when compared to fine-resolution simulations.

> To make sure it is clear that the potential for resolution-dependent biases is an important consideration when choosing a stretch-factor, we have added the following to the description of constrain 2,

*The minimum resolution, $R_{min}$, is an important consideration. $R_{min}$ imposes a limit on the coarsest resolution outside of the refined domain. Therefore, the choice of an appropriately fine $R_{min}$ can reduce potential bias in species in the coarse grid outside of the refined domain from affecting the study species in the refined domain.*

along with a new paragraph at the end of Sect. 2.3,

*The single refinement and the expansion of grid-boxes outside the refined domain are important limitations of grid-stretching. The coarse grid outside the refined domain is susceptible to resolution-dependent biases in $O_3$ and CO (Wild and Prather, 2006; Yan et al., 2014, 2016), which could influence the representation of chemical processes in the refined domain. Therefore, the choice of an appropriately fine minimum resolution is important ($R_{min}$ in constraint 2). Generally, stretched-grid simulations are well suited for applications that are principally sensitive to emissions and physical processes in the refined domain. For example, stretched-grid simulations are well suited for regional studies of boundary layer concentrations of short-lived species. Applications such as evaluations of the global tropospheric $O_3$ budget are better suited for standard cubed-sphere simulations.*

We have also added a remark to Sect. 2.5 that points this feature out,

*We note a small low bias in ozone above 400 hPa associated with nonlinear chemical sensitivity to coarse grid-boxes outside of the refined region. The C96e-NA grid resolution is coarser than 250 km in most of Asia, while the C96-global grid is finer than 125 km (Figure 4). A finer grid size or a smaller stretch-factor could be used to increase resolution outside the refinement if this type of bias is of concern for a stretched-grid application.*

[Rev. 2, comment 2.b.] With the recent improvement in computer performance, atmospheric simulations using thousands or tens of thousands of CPU cores are no longer unusual. In terms of time-to-solution, the stretch calculation of C180e-US is only 2.5 times faster than that of C180-global. The authors should consider whether they will tolerate the side effects and reduce computing resource consumption by just a few hundred cores.

In general, the majority of GEOS-Chem users are working with a few dozen to a few hundred CPUs. In this context, global fine resolution simulations are expensive, so the ability to reduce compute resources by a few hundred cores is valuable. To clarify, C180e-US was more than twice as fast as C180-global on eight times fewer cores. We have clarified this in the text:

*In terms of model throughput, C180e-US was 2.4 times faster than C180-global, despite using 8 times fewer cores.*

We agree with the reviewer that it is necessary to consider the possible effects of the coarse grid boxes outside the refined domain. Constraint 2 in Sect. 2.3 is specifically included to address this. $R_{min}$ is a mechanism to limit the coarse resolutions outside the refinement. We have clarified this in the description of Constraint 2:

*The minimum resolution, $R_{min}$, is an important consideration. $R_{min}$ imposes a limit on the coarsest resolution outside of the refined domain. Therefore, the choice of an appropriately fine $R_{min}$ can reduce potential bias in species in the* coarse grid outside of the refined domain from affecting the study species in the refined domain.

Ultimately, determining if a stretched-grid is suitable for an application is application-specific. We have added a paragraph at the end of Sect. 2.3 to ensure this is clear:

*The single refinement and the expansion of grid boxes outside the refined domain are important limitations of grid-stretching. The coarse grid outside the refined domain is susceptible to resolution-dependent biases in $O_3$ and CO (Wild and Prather, 2006; Yan et al., 2014, 2016), which could influence the representation of chemical processes in the refined domain. Therefore, the choice of an appropriately fine minimum resolution is important ($R_{min}$ in constraint 2). Generally speaking, stretched-grid simulations will be best suited for applications that are principally sensitive to emissions and physical processes in the refined domain. For example, stretched-grid simulations would be well suited for regional studies of boundary layer concentrations of short-lived species. Applications such as evaluations of the global tropospheric $O_3$ budget would be better suited for standard global simulations.*

[Rev. 2, comment 3.a.] In more aggressive stretching experiments, the advantage of high resolution should be examined from multiple perspectives. Many regional chemical transport simulations have already shown that higher resolution emission data can reproduce the tracer concentration distribution near the emission source. The authors should emphasize the advantages of the grid-stretched global chemical transport model that are not present in the regional model. For example, I want to know what advantage there is in keeping the resolution around New York at the same degree as in the non-stretch experiment.

We now clarify that we implement grid stretching in GCHP because grid refinement is a valuable capability for a global model, and previously there was no grid refinement capability.  The choice to implement stretching rather than nesting was pragmatic; currently, nested grids are not feasible to implement because of the technical complexity.

Throughout the manuscript we emphasize how stretching modifies the grid because it is necessary to understand how the resolution varies in order to determine if stretching is suitable for an application, and if it is, what range of stretch-factors are appropriate. We include the example of how to estimate C900e-CA's resolution in New York to demonstrate how one can estimate the grid resolution at a location.

[Rev. 2, comment 3.b.] Besides, the author should evaluate that they chose a higher resolution than the meteorological field. I recommend adding a stretch calculation experiment with a resolution close to that of the meteorological field. If that resolution gives almost the same results as the 10km resolution in this study's evaluation method, there is no need to use the 10km resolution.

We thank the reviewer for this suggestion. We checked that using a finer model grid than the meteorological data did not have any significant negative consequences for our application, and we included that in Appendix D. We upscaled the 0.25°x0.3125° (~25 km) meteorological data to 0.5°x0.625° (~50 km), and then conducted two simulations with ~25 km resolution in California: one with the native meteorological data, and one with the upscaled data. In the simulation with the upscaled data, we did not see any concerning effects on the simulated $NO_2$ columns. In Appendix D we also include the TROPOMI and C900e-CA columns for comparison with the ~25 km simulations.

We also updated Sect. 3 so that the purpose of the demonstrations in Sect. 3.1 and 3.2 is more clear. We state more clearly that Sect. 3.2 is intended to push the limits of grid stretching in GCHP, and to probe the idea of using a large stretch-factor for a highly localized refinement. With a standard cubed-sphere grid, ~10 km resolution is not feasible computationally; Sect. 3.2 demonstrates that stretched-grid simulations are capable of such fine resolutions.

> [In Sect. 3] we explore two of the primary reasons one might use the stretched-grid capability: (1) for a computationally efficient regional simulation, and (2) to realize a finer resolution than otherwise possible. Sect. 3.1 considers a comparison of columns in the contiguous US; columns are simulated at ~50 km resolution, with stretched-grid and standard cubed-sphere simulations, to examine the ability of a stretched-grid simulation to produce similar results to a cubed-sphere simulation at a lesser computational expense. Sect. 3.2 experiments with the use of a very large stretch-factor, for a simulation targeting California with ~10 km resolution and modest computational requirements.

**Specific comments**

**Section 1: Introduction**

[Rev. 2, comment 4.] Although it is mentioned in Harris et al.(2016), Tomita(2008) should be referred to, as a previous study of global atmospheric models with stretch grids using the Shumidt transform. The studies of Goto et al.(2015) and Trieu et al.(2017), which simulated atmospheric chemical transport on the stretched grid, should also be cited.

We thank the reviewer for pointing this out. We have added references to Tomita (2008), Goto et al. (2015), and Trieu et al. (2017), as well as references to Allen et al. (2000), Park et al. (2005), McGregor and Dix (2008), Uchida et al. (2016).

[Rev. 2, comment 5.] Lines 38-46: It seems to me that almost all the tools to realize stretch calculations were available before this study, and there are no technical problems to be solved to perform chemical transport calculations on a stretched-grid. I would like to see a clarification of what the authors contributed to the implementation.

We now clarify that our contribution is the practical work that was involved in going from the capabilities of individual components, to a unified and validated model capability. More specifically, testing that the individual capabilities worked together (and debugging issues), upgrading model components, and developing the utilities

for conducting stretched-grid simulations, which are now available to the community. Our manuscript describes this model improvement, and considerations for using stretched-grid simulations.

[Rev. 2, comment 6.] Lines 65-66: Is the meteorological field re-gridded for both horizontal and vertical, and if so, how is the transformation applied between model grids with different topography?

We thank the reviewer for pointing out that this is not clear. We have updated the model description in Sect. 2.1, and include an expanded description of the horizontal grid and regridding, the vertical grid, and how the effects of topography are handled.

*Emissions and meteorological input data are regridded online by ESMF using the first-order conservative scheme originally described in Ramshaw (1985)…*

*...The effect of topography and surface type on transport is implicit in the meteorological data that drives transport. Both data products have a 72-level terrain-following hybrid-sigma pressure grid that extends from the surface to 1 Pa. GCHP uses a vertical grid that is identical to meteorological data, so vertical regridding is not required.*

[Rev. 2, comment 7.] Lines 66-67: Does the model solve the stratospheric chemical processes, and if not, how is the stratospheric gas (ozone) concentration given?

The model solves stratospheric chemical processes using the Unified Chemistry Extension (UCX), described in Eastham et al., (2014). We added a phrase describing this in Sect. 2.1.

*Stratospheric chemistry is simulated using the Unified Chemistry Extension (UCX) described in Eastham et al. (2014).*

[Rev. 2, comment 8.] Lines 135-136: The number of vertical layers and each layer's altitude in the meteorological field is the same as in the GEOS-Chem settings? How much higher is the top altitude for the meteorological field than for the GEOS-Chem simulations?

The vertical grid is a hybrid sigma-pressure grid (terrain following with a gradual transition to a pressure grid) that extends from the surface to 1 Pa. The vertical grid is identical to the vertical grid of the meteorological data. We have updated Sect. 2.1 to include a description of the vertical grid and a remark that the vertical grid is the same as the meteorological data.

*Both [meteorological] data products have a 72-level terrain-following hybrid-sigma pressure grid that extends from the surface to 1 Pa. GCHP uses a vertical grid that is identical to meteorological data, so vertical regridding is not required.*

[Rev. 2, comment 9.] Lines 136-137: For which time period was this simulation performed?

The configuration described in Sect. 2.4 (includes lines 136--137) is a shared configuration for all the simulations we conducted for this manuscript (C96-global,

C94-global, C96e-NA, C180-global, C180e-US, C60-global, C900e-CA, C90-global). The specific simulation periods are mentioned in the respective sections.

The simulations in Sect. 2.5 ran from June through September 2018, and the mean concentrations for September are compared. We have clarified this in the text:

> *These comparisons are for the fourth simulation month, to accommodate relaxation time for CO and $O_3$ (the simulations started June 1, 2018 and ran through October 1, 2018).*

The simulations in Sect. 3.1 and 3.2 ran from June through July 2018, and the mean column densities for July 2018 are shown; however, in response to Rev. 1, comment 29 we changed the simulations in Sect. 3.1 and 3.2 to run from June through July 2019. We now clarify the simulation periods in Sect. 3.1:

> *Figure 7 shows tropospheric $NO_2$ column densities from TROPOMI, C180-global, C180e-US, and C60-global for the US in July 2019. All three simulations included a 1-month spinup.*

and in Sect. 3.2:

> *Figure 9 shows tropospheric $NO_2$ column densities from TROPOMI, C900e-CA, and C90-global for California in July 2019. Both simulations used a 1-month spinup.*

[Rev. 2, comment 10.] Lines 154-155: I want to know why the authors did not align the Face center of the hexahedron between the focal region of the C96e-NA and the C96-global.

> The cubed-sphere faces are fixed in GCHP, so the center of a face cannot be aligned to a point. We note that a stretched-grid with S=1 would be capable of this, but for this test, it was important that the stretching capability was disabled in the reference simulations (C96-global, and C94-global).
>
> We have expanded our description of the cubed-sphere grid in Sect. 2.1 and added a note that the cubed-sphere faces are fixed.
>
> > *Each face is a logically square grid that is regularly spaced in a gnomonic projection centered on the face. One of the six faces is highlighted in Figure 1. The position of the faces are fixed, and the center of the first face is 0° N, 10° W .*

[Rev. 2, comment 11.a.] Lines 163-164, Figure 4: 1) In order to clarify the difference in the free troposphere, please show the results for the layer near the surface and the layer above the surface separately.

> Done.

[Rev. 2, comment 11.b.] ... 2) As mentioned in the major comment, the difference in ozone concentration in the free troposphere (left panel) appears to be biased rather than scattered. I would like to see further analysis of this.

We thank the reviewer for this comment. We have added a paragraph to the end of Sect. 2.3 (choosing an appropriate stretch factor) that makes the following points:

- The single refinement, and coarsening of grid boxes outside the refinement are an inherent limitation of stretching. These make stretched-grid simulations susceptible to resolution dependent biases in $O_3$ and CO, like those described in Wild and Prather (2006), Yan et al., (2014), and Yan et al., (2016).
- We reinforce that for a given application, a choice of an appropriately fine minimum resolution is an important consideration.
- We note that stretched-grid simulations are likely best suited for applications that are primarily sensitive to modes of variability in the target face.

There is a small low bias in ozone above 350--400 hPa associated with the coarse grid-boxes outside of the refined domain. The C96e-NA grid is coarser than 250 km in most of Asia, while the C96-global grid is finer than 125 km. The small low bias is consistent with the general tendency of coarse simulations to underestimate $O_3$ concentrations aloft when they are compared to fine-resolution simulations.

Sect. 2.5 is focused on validating the technical implementation of stretched-grid, so we do not find this concerning. We have added a remark to Sect. 2.5 detailing this.

*We note a small low bias in ozone above 400 hPa. This is associated with nonlinear chemical sensitivity to coarse grid-boxes outside of the refined region. The C96e-NA grid resolution is coarser than 250 km in most of Asia, while the C96-global grid is finer than 125 km (Figure 4). A finer grid size or a smaller stretch-factor could be used to increase resolution outside the refinement if this type of bias is of concern for a stretched-grid application.*

To demonstrate here, we reran C96e-NA with a larger grid size and a smaller stretch-factor so that the coarse grid outside the refinement was finer (a finer $R_{min}$ by constraint 2) but the target resolution was not changed (C96e). Here, we refer to this rerun simulation as C96e-NA2. The stretch-factor was 1.6 and the grid size was C72; this means the grid outside the refinement was more than twice as fine as it was in C96e-NA (but still more than twice as coarse as C96-global in places). Below is the scatter plot for ozone for C96e-NA2 versus C96-global, and it shows that the bias in ozone above 400 hPa is gone.

[Figure]

[Rev. 2, comment 11.c.] … 3) In the figure for OH, the colors are largely different between the right and left figures. In the left figure, where did the red dot around 0.1-0.2ppt go?

The right column compares C94-global and C96-global and gives an expectation for the amount of differences we should expect from aliasing-artifacts between two grids with similar resolution (it is an estimate of precision). The overlap of C94-global and C96-global grid-boxes has a different spatial pattern than the overlap of C96e-NA and C96-global grid-boxes; therefore, we should expect some differences between the left and right scatter plots. The differences in the left and right scatter plots are generally consistent, and thus demonstrate that the technical implementation is sound.

The "wisps" in the OH scatter plots are columns where the source grids (C96e-NA and C94-global) do not overlap well with the destination grid (C96-global). We can illustrate this by changing the colormap to show the maximum regridding weight (maximum fraction of overlap between the destination grid-box and source grid-box; 1=perfect overlap, 0.25=destination grid-box evenly split between 4 source grid-boxes):

[Figure]

[Rev. 2, comment 12.a.] Lines 204-205, Table2: 1) The differences in the number of grid points and the number of time steps (time intervals) between the C180-global and C60-global experiments are not described. Since the relative difference in computational workload can be approximated by the number of grid points and the number of time steps, this information is necessary to evaluate the computational performance.

We have added an expanded description of the cubed-sphere grid in Sect. 2.1 (model description) and include a table comparing different cubed-sphere grids. This table includes the total number of grid boxes in a C60 grid and a C180 grid. We also added a remark to Sect. 2.1 that the total number of grid boxes is a reasonable approximation of the computational workload.

*The computational demand of a GCHP simulation is proportional to the total number of grid-boxes. Table 1 provides a comparison of cubed-sphere grids and conventional latitude-longitude grids.*

In Sect. 3.1 (includes lines 204--205) we have clarified that C60-global is effectively C180e-US with stretching disabled.

> *To examine C180e-US without grid-stretching, we conduct a third simulation, C60-global.*

[Rev. 2, comment 12.b.] … It is not stated how many months of calculation results were used for the comparison. Recently, the unit "Simulation Year Per wall clock Day (SYPD)" is often used to express the elapsed time of a simulation. This should be added.

> On line 196--197 (now line 252--253) we mention that the comparison is for July 2018 and that all three simulations used a 1-month spinup. The caption of Figure 6 also states that it is showing the mean tropospheric $NO_2$ column densities for July 2018.
>
> We thank the reviewer for suggesting the addition of a model throughput. We have added this to the table.

[Rev. 2, comment 13.] Lines 224-225, Figure 8: Some people don't even know which is north for LA or SF. I don't know which area is the Mojave Desert. Please add a mark to the figure.

> We have added an annotated map of California in Appendix C (Figure C2). In the text and in the caption we refer the reader to Figure C2 for the annotated map.

[Rev. 2, comment 14.a.] Lines 231-233: 1) I interpreted that the authors upscaled the 0.25degx0.3125deg meteorological data to a grid resolution of ~100 km in the C90-global experiment and interpolated it to a grid resolution of ~10 km in the C900e-CA. Does this mean that a spatial resolution of 0.25degx0.3125deg is sufficient to represent the concentration gradient around LA?

> The reviewer's interpretation of upscaling/downscaling of the meteorological data is correct. Our use of 0.25°x0.3125° meteorological data is a practical limitation, as it is the finest meteorological data currently available for GCHP. In the model description (Sect. 2.1) we have clarified that the 0.25°x0.3125° GEOS-FP data product is the finer of the two meteorological data products currently available for GCHP:
>
> > *Offline meteorological data for GCHP are provided by the GMAO. GCHP uses a local archive of the GEOS-FP or MERRA-2 data product. GEOS-FP is a near real-time analysis product with a 0.25°×0.3125° grid. MERRA-2 is a reanalysis product with a 0.5°×0.625° grid.*
>
> We do expect 0.25°x0.3125° resolution is too coarse to resolve some of the orographic and coastal effects in California; however, it is the finest resolution data currently available. We have added a remark about this to Sect. 3.2:
>
> > *C900e-CA uses meteorological data from the GEOS-FP data product with a spatial resolution of 0.25°×0.3125° (~25 km). We expect some of the detailed orographic and coastal effects in California to be missed. C900e-CA identifies a need for even finer resolution meteorological data, for which there is ongoing work in the GCHP–GMAO community.*

[Rev. 2, comment 14.b.] … 2) To discuss why the concentration gradients were well reproduced, it is necessary to evaluate whether the higher resolution of the emission or the higher resolution of the meteorological field was more effective. Previous researches should

be presented. Or sensitivity experiments should be conducted, such as using up scaled, low-resolution emission inventories or meteorological fields in high-resolution experiments.

> We have rephrased the overview and description of the case studies in Sect. 3 so that their purpose is more clear. The case study in Sect. 3.2 is intended to demonstrate and explore a stretched-grid simulation with a large stretch-factor, and to demonstrate that stretched-grid simulations are easily capable of resolutions that are significantly finer than previously possible with GCHP.

> This manuscript is a model development type manuscript describing and demonstrating the new stretched-grid capability of GCHP; therefore, an evaluation of the relative importance of emission versus meteorological data resolution on simulated processes in California beyond the scope of this manuscript.

> We do note that Appendix D considers the effects of upscaling the 0.25°x0.3125° (~25 km) meteorological data to 0.5°x0.625° (~50 km), and whether that has a significant effect on the simulated $NO_2$ columns. The differences are small, which suggests that emissions are driving much of the variability seen in C900e-CA; this makes sense given the emissions data we use has ~9 km resolution, and since $NO_x$ lifetime is short.

[Rev. 2, comment 14.c.] … 3) GEOS-Chem is a transport model, and the wind field is a given. I want the authors to show the benefits of usage of higher resolution than the meteorological field for vertical transport, diffusion and chemical reaction in detail. The 10km-mesh simulation provides a better representation of the concentration gradient near the emission source than the 0.25deg-mesh? I would like to see an analysis of the ozone production/dissipation balance, etc.

> We thank the reviewer for their comment. Similar to the previous comment, we feel that the purpose of Sect. 3.2 was perhaps misunderstood. The case study in Sect. 3.2 is intended to demonstrate an application that could be suited for a stretched-grid simulation with a large stretch-factor, and to demonstrate that stretched-grid simulations are capable of very fine resolution (in terms of a global model).

> We reiterate that we use 0.25°x0.3125° meteorological inputs because that is the finest resolution meteorological data that was available for GCHP. The C900e-CA simulation demonstrates that the model is now capable of finer resolution than the meteorological data, which motivates the need for further work to develop finer resolution meteorological inputs (work which is underway).

> Future work such as evaluating the effects of model resolution and input data resolution on transport processes and chemical processes, or a model evaluations of GCHP at fine resolution in regions like California, would indeed be interesting studies which stretched-grid simulations could be useful for. We focus this manuscript on describing and demonstrating the stretched-grid model improvement in GCHP version 13.0.0.

> We have rephrased the description of the case studies in Sect 3 to make their purpose more clear.

*Next, [in Sect. 3,] we demonstrate stretched-grid simulations with GCHP by conducting case studies in Sect. 3.1 and 3.2. The applications we consider are regional comparisons of simulated tropospheric $NO_2$ columns with observations. $NO_2$ is chosen because it is a well-measured species, and the sensitivity of its simulated concentrations to model resolution and local chemical and physical processes make it a prime example of an application that is well-suited for a stretched-grid simulation. Here we explore two of the primary reasons one might use the stretched-grid capability: (1) for a computationally efficient regional simulation, and (2) to realize a finer resolution than otherwise possible. Sect. 3.1 considers a comparison of columns in the contiguous US; columns are simulated at ~50 km resolution, with stretched-grid and standard cubed-sphere simulations, to examine the ability of a stretched-grid simulation to produce similar results to a cubed-sphere simulation at a lesser computational expense. Sect. 3.2 experiments with the use of a very large stretch-factor, for a simulation targeting California with ~10 km resolution and modest computational requirements.*

**References**

Allen, D., Pickering, K., Stenchikov, G., Thompson, A., and Kondo, Y.: A three-dimensional total odd nitrogen (NO y) simulation during SONEX using a stretched-grid chemical transport model, Journal of Geophysical Research: Atmospheres, 105, 3851–3876, 2000.

Eastham, S. D., Weisenstein, D. K., and Barrett, S. R.: Development and evaluation of the unified tropospheric–stratospheric chemistry extension (UCX) for the global chemistry-transport model GEOS-Chem, Atmospheric Environment, 89, 52–63, https://doi.org/10.1016/j.atmosenv.2014.02.001, https://linkinghub.elsevier.com/retrieve/pii/S1352231014000971, 2014.

Freitas, S. R., Grell, G. A., Molod, A., Thompson, M. A., Putman, W. M., Santos e Silva, C. M., & Souza, E. P. (2018). Assessing the Grell-Freitas convection parameterization in the NASA GEOS modeling system. Journal of Advances in Modeling Earth Systems, 10, 1266– 1289. https://doi.org/10.1029/2017MS001251

Freitas, S. R., Putman, W. M., Arnold, N. P., Adams, D. K., & Grell, G. A. (2020). Cascading toward a kilometer-scale GCM: Impacts of a scale-aware convection parameterization in the Goddard Earth Observing System GCM. Geophysical Research Letters, 47, e2020GL087682. https://doi.org/10.1029/2020GL087682

Goto, D., Dai, T., Satoh, M., Tomita, H., Uchida, J., Misawa, S., Inoue, T., Tsuruta, H., Ueda, K., Ng, C., et al.: Application of a global nonhydrostatic model with a stretched-grid system to regional aerosol simulations around Japan, Geoscientific Model Development, 8, 235–259, 2015.

McGregor, J. L. and Dix, M. R.: An updated description of the conformal-cubic atmospheric model, in: High resolution numerical modelling of the atmosphere and ocean, pp. 51–75, Springer, 2008.

Park, R., Pickering, K., Allen, D., Stenchikov, G., and Fox-Rabinovitz, M.: Global simulation of tropospheric ozone using the University of Maryland Chemical Transport Model (UMD-CTM): 2. Regional transport and chemistry over the central United States using a stretched grid, Journal of Geophysical Research: Atmospheres, 109, 2004.

Ramshaw, J. D.: Conservative rezoning algorithm for generalized two-dimensional meshes, Journal of Computational Physics, 59, 193–199, https://doi.org/10.1016/0021-9991(85)90141-X, https://linkinghub.elsevier.com/retrieve/pii/002199918590141X, 1985.

Tomita, H.: A stretched icosahedral grid by a new grid transformation, Journal of the Meteorological Society of Japan. Ser. II, 86, 107–119, 2008.

Trieu, T. T. N., Goto, D., Yashiro, H., Murata, R., Sudo, K., Tomita, H., Satoh, M., and Nakajima, T.: Evaluation of summertime surface ozone in Kanto area of Japan using a semi-regional model and observation, Atmospheric Environment, 153, 163–181, 2017.

Uchida, J., Mori, M., Nakamura, H., Satoh, M., Suzuki, K., and Nakajima, T.: Error and energy budget analysis of a nonhydrostatic stretched-grid global atmospheric model, Monthly Weather Review, 144, 1423–1447, 2016.

Valin, L. C., Russell, A. R., Hudman, R. C., and Cohen, R. C.: Effects of model resolution on the interpretation of satellite NO 2 observations, Atmospheric Chemistry and Physics, 11, 11 647–11 655, https://doi.org/10.5194/acp-11-11647-2011, https://www.atmos-chem-phys.net/11/11647/2011/, 2011.

Wild, O. and Prather, M. J.: Global tropospheric ozone modeling: Quantifying errors due to grid resolution, Journal of Geophysical Research, 111, D11 305, https://doi.org/10.1029/2005JD006605, http://doi.wiley.com/10.1029/2005JD006605, 2006.

Yan, Y., Lin, J., Chen, J., and Hu, L.: Improved simulation of tropospheric ozone by a global-multi-regional two-way coupling model system, Atmospheric Chemistry and Physics, 16, 2381–2400, https://doi.org/10.5194/acp-16-2381-2016, https://www.atmos-chem-phys.net/16/2381/2016/, 2016.

Yan, Y.-Y., Lin, J.-T., Kuang, Y., Yang, D., and Zhang, L.: Tropospheric carbon monoxide over the Pacific during HIPPO: two-way coupled simulation of GEOS-Chem and its multiple nested models, Atmospheric Chemistry and Physics, 14, 12 649–12 663, https://doi.org/10.5194/acp-14-12649-2014, https://www.atmos-chem-phys.net/14/12649/2014/, 2014.

Zängl, G., Reinert, D., Rípodas, P., and Baldauf, M.: The ICON (ICOsahedral Non-hydrostatic) modelling framework of DWD and MPI-M: Description of the non-hydrostatic dynamical core, Quarterly Journal of the Royal Meteorological Society, 141, 563–579, 2015.